# TRPC3 is a major contributor to functional heterogeneity of cerebellar Purkinje cells

Bin Wu[1], François GC Blot[1], Aaron Benson Wong[1], Catarina Osório[1], Youri Adolfs[2], R Jeroen Pasterkamp[2], Jana Hartmann[3], Esther BE Becker[4], Henk-Jan Boele[1], Chris I De Zeeuw[1,5], Martijn Schonewille[1]*

[1]Department of Neuroscience, Erasmus Medical Center, Rotterdam, Netherlands; [2]Department of Translational Neuroscience, University Medical Center Utrecht, Utrecht University, Utrecht, Netherlands; [3]Institute of Neuroscience, Technical University Munich, Munich, Germany; [4]Department of Physiology, Anatomy and Genetics, University of Oxford, Oxford, United Kingdom; [5]Netherlands Institute for Neuroscience, Royal Dutch Academy for Arts and Sciences, Amsterdam, Netherlands

**Abstract** Despite the canonical homogeneous character of its organization, the cerebellum plays differential computational roles in distinct sensorimotor behaviors. Previously, we showed that Purkinje cell (PC) activity differs between zebrin-negative (Z–) and zebrin-positive (Z+) modules (Zhou et al., 2014). Here, using gain-of-function and loss-of-function mouse models, we show that transient receptor potential cation channel C3 (TRPC3) controls the simple spike activity of Z–, but not Z+ PCs. In addition, TRPC3 regulates complex spike rate and their interaction with simple spikes, exclusively in Z– PCs. At the behavioral level, TRPC3 loss-of-function mice show impaired eyeblink conditioning, which is related to Z– modules, whereas compensatory eye movement adaptation, linked to Z+ modules, is intact. Together, our results indicate that TRPC3 is a major contributor to the cellular heterogeneity that introduces distinct physiological properties in PCs, conjuring functional heterogeneity in cerebellar sensorimotor integration.
DOI: https://doi.org/10.7554/eLife.45590.001

*For correspondence:
m.schonewille@erasmusmc.nl

Competing interests: The authors declare that no competing interests exist.

## Introduction

Maintaining correct sensorimotor integration relies on rapid modifications of activity. The cerebellum is instrumental herein, evidenced by the fact that disruptions of cerebellar functioning, for example through stroke or neurodegenerative disorders, affect coordination and adaptation of many types of behaviors such as gait, eye movements and speech (*Ackermann et al., 1992*; *Bodranghien et al., 2016*). The palette of behavioral parameters controlled by the cerebellum is also broad and includes features like timing (*Raymond et al., 1996*; *De Zeeuw and Yeo, 2005*; *Yang and Lisberger, 2014*), strength (*Hirata and Highstein, 2000*; *Witter et al., 2013*), as well as coordination of muscle activity (*Thach et al., 1992*; *Vinueza Veloz et al., 2015*). However, the pluriformity of behavioral features does not match with the homogeneity of the structure and cyto-architecture of the cerebellar cortex.

Recently, it has been uncovered that the sole output neurons of the cerebellar cortex, the Purkinje cells (PCs), can be divided into two main groups with a distinct firing behavior (*Xiao et al., 2014*; *Zhou et al., 2014*). One group, consisting of PCs that are positive for the glycolytic enzyme aldolase C, also referred to as zebrin II (*Brochu et al., 1990*; *Ahn et al., 1994*), shows relatively low simple spike firing rates, whereas the PCs in the other group that form zebrin-negative zones, fire at higher rates (*Zhou et al., 2014*). Zebrin II demarcates olivocerebellar modules, anatomically defined operational units each consisting of a closed loop between the inferior olive, parasagittal bands of

the cerebellar cortex and the cerebellar nuclei (*Sugihara and Quy, 2007*; *Ruigrok, 2011*). Given that different motor domains are controlled by specific olivocerebellar modules (*Horn et al., 2010*; *Ruigrok, 2011*; *Graham and Wylie, 2012*), the differential intrinsic firing frequencies may be tuned to the specific neuronal demands downstream of the cerebellum (*De Zeeuw and Ten Brinke, 2015*). Thus, dependent on the specific behavior controlled by the module involved, the PCs engaged may show low or high intrinsic firing as well as related plasticity rules to adjust these behaviors (*Apps et al., 2018*).

Cellular heterogeneity can drive differentiation in the activity and plasticity of individual cells that operate within a larger ensemble (*Altschuler and Wu, 2010*). The molecular and cellular determinants of differential electrophysiological processing in the cerebellar PC modules are just starting to be identified (*Cerminara et al., 2015*; *Apps et al., 2018*). For example, while the impact of zebrin II itself is still unclear (*Zhou et al., 2014*), excitatory amino acid transporter 4 (EAAT4) and GLAST/ EAAT1 may selectively modulate simple spike activity of zebrin-positive PCs as well as plasticity of their parallel fiber (PF) inputs (*Wadiche and Jahr, 2005*; *Perkins et al., 2018*). Likewise, the distributions of particular subcategories of receptors that may be relevant for firing properties are linked to the same modular organization. For example, whereas γ-aminobutyric acid type B (GABA$_B$) receptors occur in both zebrin-positive and zebrin-negative PCs (*Tian and Zhu, 2018*), the GABA$_{B2}$ receptor is selectively expressed in a pattern similar to that of zebrin II (*Chung et al., 2008*). Or, whereas the alpha isoform of mGluR1 (mGluR1a) is uniformly expressed in all PCs (*Ohtani et al., 2014*) the mGluR1b receptor is expressed in a pattern complementary to that of zebrin II (*Mateos et al., 2001*). Interestingly, the modular distributions of most of these receptors point towards a critical role of transient receptor potential cation channel subfamily C member 3 (TRPC3) in regulating electrophysiological properties of PCs. For example, while the mGluR1b receptor interacts with TRPC3 to drive mGluR1-dependent currents (*Hartmann et al., 2008*), the GABA$_B$ receptors modulate mGluR1-triggered TRPC3-mediated currents (*Tian and Zhu, 2018*). However, where and how TRPC3 operates in cerebellar PCs is still largely unknown (*Zhou et al., 2014*).

Here, we set out to test the hypothesis that TRPC3 is a key player in the molecular machinery responsible for differential control over the activity and function of Z+ and Z– PCs. We demonstrate that TRPC3 in the brain has particularly high expression levels in the cerebellum, in a pattern largely, but not precisely, complementary to zebrin. We examined the impact of TRPC3 gain-of-function and loss-of-function mutations and found effects on the spiking rate of Z– but not Z+ PCs in vitro. In vivo recordings during quiet wakefulness in the same mutants revealed that the level of TRPC3 influences both simple spike and complex spike rates, and the interaction between the two, also selectively in Z– modules. Finally, we show that adaptation of compensatory eye movements, which is controlled by Z+ modules in the vestibulocerebellum (*Sanchez et al., 2002*; *Zhou et al., 2014*), is not affected by the loss of TRPC3 function, whereas the learning rate during eyeblink conditioning, which is linked to the Z– modules (*Hesslow, 1994*; *Mostofi et al., 2010*), is decreased after PC-specific ablation of TRPC3, highlighting the behavioral relevance of firing rate modulation by TRPC3.

## Results

### Specific expression pattern and subcellular localization of TRPC3 in the mouse brain

As the expression of TRPC3 in the adult mammalian brain is still unclear, we first set out to examine the immunohistochemistry of TRPC3 using a novel TRPC3-specific antibody (Cell signaling, #77934). We found that in the normal mouse brain TRPC3 is most prominently expressed in the olivocerebellar circuit (*Figure 1A*), specifically in PCs and unipolar brush cells (UBCs) (*Figure 1B*). This is in line with previous immunostainings and in situ data (Allen Brain Atlas, http://mouse.brain-map.org/). Upon further scrutiny it is clear that, although expressed in all PCs, endogenous TRPC3 was not distributed homogeneously. The TRPC3 levels in the anterior cerebellum, where the PCs are predominantly Z–, were higher than those in the posterior PCs, which are primarily Z+ (*Figure 1A*, *Figure 1—figure supplement 1*). To further visualize the relationship between TRPC3 and Zebrin II, labeled as Aldolase C (*Ahn et al., 1994*), we quantified their relative levels in several subregions

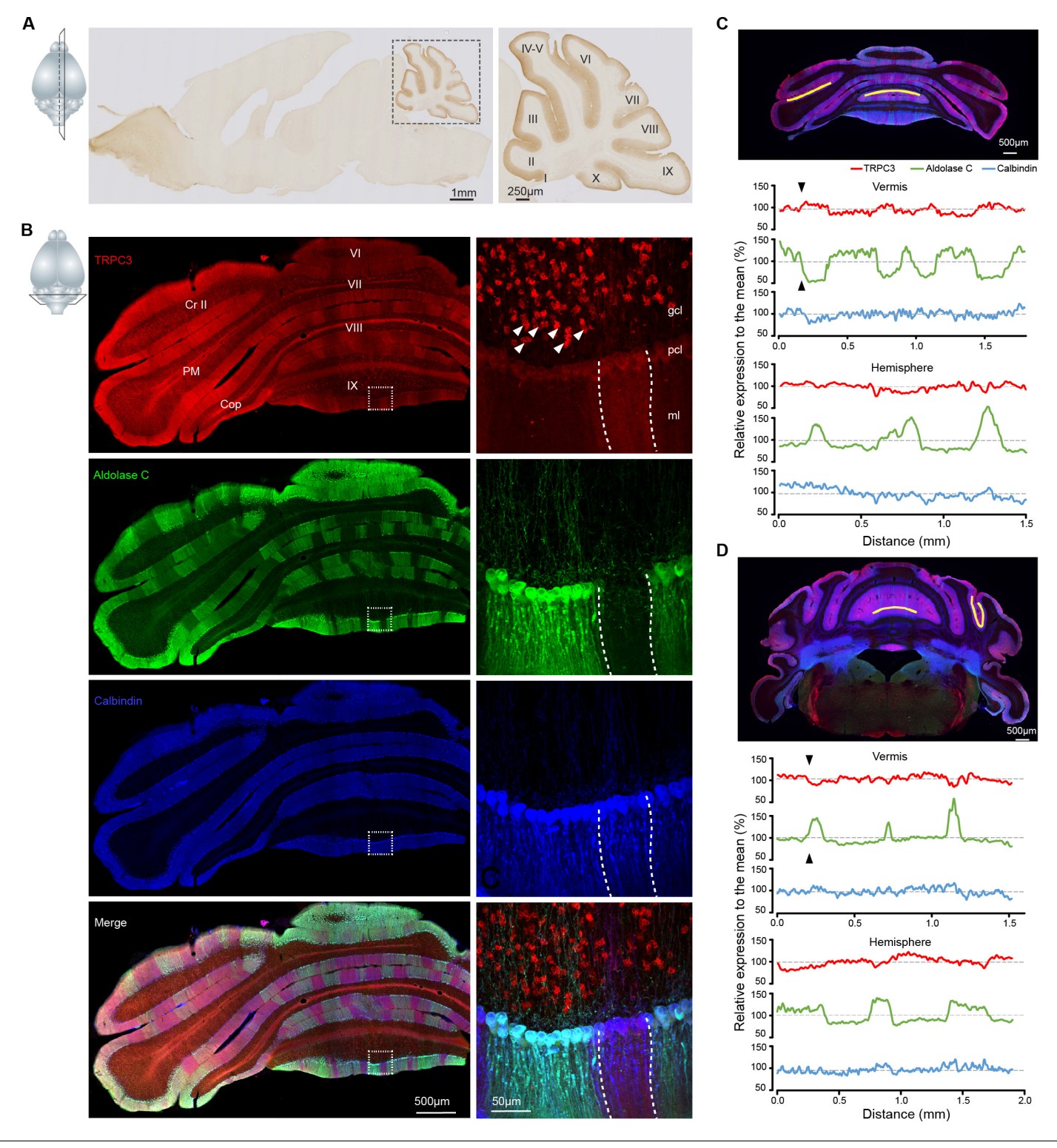

**Figure 1.** TRPC3 is predominantly expressed in the cerebellum in a zebrin-related pattern. (A) Representative image and magnification (right) of sagittal cryosection of an adult mouse brain stained with anti-TRPC3. Inset, plane of section. (B) Coronal immunofluorescence images with anti-TRPC3 (red), anti-Zebrin II/Aldolase C (green) and anti-calbindin (blue) staining of the cerebellar cortex (left), with magnifications (right). TRPC3 is expressed in the cerebellar PCs and UBCs (triangles), in a pattern that in the vermis complements that of zebrin and appears more uniform in the hemispheres. Inset, plane of section. (C) Posterior coronal section of the cerebellar cortex (top) used to performed a quantification of the relative intensity of immunofluorescence staining of TRPC3, Zebrin II/Aldolase C and calbindin for PCs in the vermis (ventral lob. VIII, middle) and the hemisphere (ventral

*Figure 1 continued on next page*

*Figure 1 continued*

PM, bottom) (values normalized to the respective means). (**D**) Similar analysis of dorsal lob. III (middle) and sulcus of Sim to Crus I (bottom) in anterior section (top). TRPC3 expression is largely complementary to Zebrin II in the vermis and parts of the hemispheres (black arrow heads), but more uniform in other hemispheric areas. In general, TRPC3 expression demonstrates a weaker differentiation between low and high levels than Zebrin II. I-X, cerebellar lobules I-X; Sim, Simplex lobule; Cr II, Crus II; PM, paramedian lobule; Cop, Copula Pyramidis; gcl, granule cell layer; pcl, Purkinje cell layer; ml, molecular layer; D, dorsal; V, ventral; M, medial; L, lateral.

DOI: https://doi.org/10.7554/eLife.45590.002

The following video and figure supplements are available for figure 1:

**Figure supplement 1.** Overview and local patterns of TRPC3 expression.

DOI: https://doi.org/10.7554/eLife.45590.003

**Figure supplement 2.** Quantification of TRPC3 expression compared to Zebrin II and calbindin.

DOI: https://doi.org/10.7554/eLife.45590.004

**Figure supplement 3.** Western blot and immunostaining of $pcp2^{Cre};TRPC3^{fl/fl}$ mice.

DOI: https://doi.org/10.7554/eLife.45590.005

**Figure 1— video 1.** Light sheet imaging reconstruction of whole-mount immunolabeling for TRPC3 (white signal), cleared with iDISCO protocol and scanned in the horizontal plane of an adult mouse brain from dorsal to ventral (see Materials and methods).

DOI: https://doi.org/10.7554/eLife.45590.006

(*Figure 1C–D* and *Figure 1—figure supplement 2*). This analysis confirmed that TRPC3 expression is complementary to that of Zebrin in the vermis, while in the hemispheres the TRPC3 expression varies between homogeneous expression and expression complementary to Zebrin II. To visualize this pattern in a more comprehensive manner, we also employed whole-mount brain light sheet imaging following iDISCO-based clearing (*Figure 1*-video 1). The antibody staining appears to be of better quality in the iDISCO protocol, resulting for instance in a clearer picture of the expression of TRPC3 in the inferior olive (most ventral, *Figure 1*-video 1). The anterior/posterior differences in the protein amount were confirmed by western blot analysis (*Figure 1—figure supplement 3A–B*).

Our immunohistochemical imaging reveals that TRPC3 is present in the soma and dendritic arbor of PCs (*Figure 1B* and *Figure 1—figure supplement 1B–E*). To further examine the subcellular localization of TRPC3 in the cerebellum, we performed immunoblots of isolated fractions following a synaptic protein extraction procedure (*Figure 1—figure supplement 3C*). As expected, TRPC3, a channel protein, is abundantly present in the membrane and almost completely absent in the cytosol (*Figure 1—figure supplement 3D*). Moreover, TRPC3 is enriched in synapstosomes (*Figure 1—figure supplement 3D*), in line with the common conception of mGluR1b-dependent activation of TRPC3 (*Hartmann et al., 2008*; *Ohtani et al., 2014*). Together, these results indicate that, within the brain, high TRPC3 expression levels are restricted to the olivocerebellar circuit, where it is present in all PCs and UBCs, but at particularly high levels in Z– PCs.

## TRPC3 differentially controls the physiological properties of PCs in vitro

Next, we investigated the contribution of TRPC3 to cerebellar function in Z+ and Z– PCs using both gain-of-function and loss-of-function mouse models (*Figure 2A*). TRPC3-Moonwalker (*TRPC3^{Mwk/-}*) mice harbor a point mutation resulting in TRPC3 gain-of-function through increased $Ca^{2+}$ influx upon activation (*Becker et al., 2009*). These mice are featured by neurodegeneration, first of UBCs and later also of PCs, and as a consequence display early onset ataxia (*Sekerková et al., 2013*). Inversely, we generated a PC-specific loss-of-function mouse model for TRPC3 (*pcp2^{Cre};TRPC3^{fl/fl}*) by crossing mice carrying *loxP*-flanked TRPC3 alleles (*Hartmann et al., 2008*) with L7-Cre (*Pcp2*-Cre) (*Barski et al., 2000*) mice. These *pcp2^{Cre};TRPC3^{fl/fl}* mice exhibited no overt signs of ataxia or other movement deficits upon visual inspection. Western blotting and immunostaining of the anterior (Z–) and the posterior (Z+) cerebellar cortex of *pcp2^{Cre};TRPC3^{fl/fl}* mice confirmed that TRPC3 protein levels are reduced, without disrupting the typical zebrin staining pattern (WB, anterior: $t_{19}$=2.63, p=0.034; posterior: $t_{19}$ = 2.67, p=0.028) (*Figure 1—figure supplement 3A–B,E–F*). The loss of TRPC3 was specific for cerebellar PCs, as TRPC3 expression in UBCs was not affected (*Figure 1—figure supplement 3F*, white arrow heads).

PCs are intrinsically active pace-making neurons, which fire regular action potentials even when deprived of synaptic inputs (*Raman and Bean, 1999*; *Womack and Khodakhah, 2002*). To

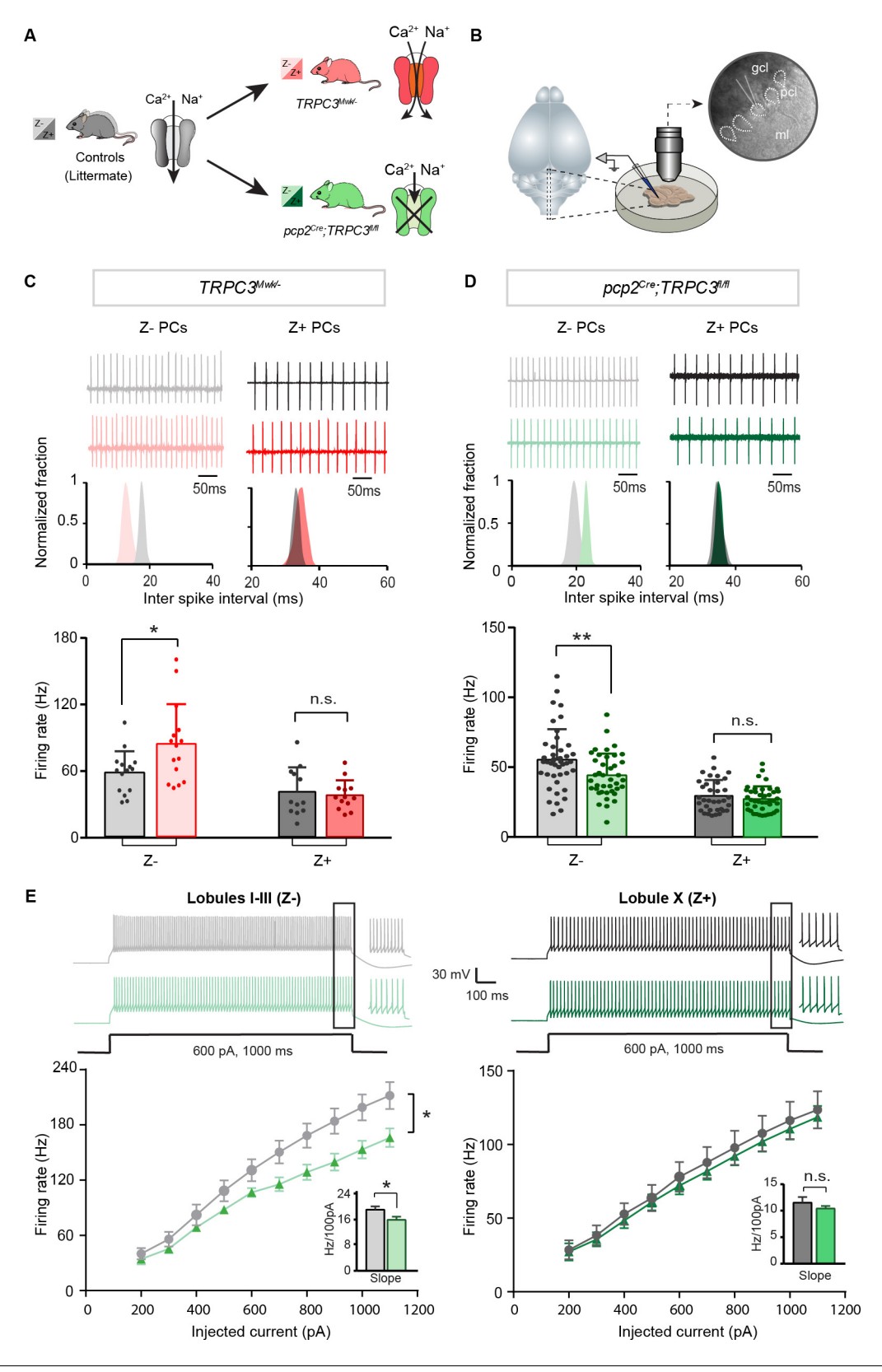

**Figure 2.** Differential controls of PC firing properties by TRPC3 in vitro. (**A**) Schematic drawing of TRPC3 channel function in control (black), gain-of-function (*TRPC3^Mwk/-^*, red) and loss-of-function (*pcp2^Cre^;TRPC3^fl/fl^*, green) mice. *Figure 2 continued on next page*

*Figure 2 continued*

(B) Schematic approach illustrating of PCs (right circle, dashed lines) recording in vitro, in acute sagittal slices. (C, D) Representative traces of cell-attached PC recordings (top) and corresponding inter spike interval (ISI) distributions (middle) in a Z– PC (left) and a Z+ PC (right) of *TRPC3*$^{Mwk/-}$ (C) and *pcp2*$^{Cre}$;*TRPC3*$^{fl/fl}$ (D) mice. Z– PCs were affected in *TRPC3*$^{Mwk/-}$ (C), light-red, n = 15 cells/N = 4 mutant mice vs. n = 15 cells/N = 4 littermate controls, $t_{28}$ = −2.47, p=0.020 and in *pcp2*$^{Cre}$;*TRPC3*$^{fl/fl}$ mice (D), light-green, n = 40/N = 6 mutants vs. n = 43/N = 5 controls, $t_{81}$ = 2.69, p=0.009). No differences in the firing rate of Z+ PCs in *TRPC3*$^{Mwk/-}$ (C), dark-red, n = 13/N = 4 mutants vs. n = 12/N = 4 controls, $t_{18}$ = 0.419, p=0.680) and *pcp2*$^{Cre}$;*TRPC3*$^{fl/fl}$ mice (D), dark-green, n = 36/N = 10 mutants vs. n = 35/N = 4 controls, $t_{64}$ = 0.937, p=0.352). (E) Whole-cell patch-clamp recordings in slice from PCs of *pcp2*$^{Cre}$;*TRPC3*$^{fl/fl}$ and control mice were used to test intrinsic excitability, by keeping cells at a holding potential of −65 mV and evoking action potentials by current steps of 100 pA (example, top). Top, exemplary traces evoked by current injection at 600 pA. Bottom, Input-output curves from whole-cell recordings of *pcp2*$^{Cre}$;*TRPC3*$^{fl/fl}$ mice of Z– PCs (left, n = 17/N = 5 mutants vs n = 17/N = 5 controls, $t_{32}$ = −2.20, p=0.035) and Z+ PCs (right, n = 12/N = 5 mutants vs n = 12/N = 4 controls, $t_{22}$ = −0.95, p=0.354). gcl, granule cell layer; pcl, Purkinje cell layer; ml, molecular layer. (C–D), data are represented as mean ± s.d.; (E), data are represented as mean ± s.e.m., * means p<0.05 and **p<0.01. For values see **Source data**.

DOI: https://doi.org/10.7554/eLife.45590.007

The following source data and figure supplement are available for figure 2:

**Source data 1.** Source data for *Figure 2* and supplement.
DOI: https://doi.org/10.7554/eLife.45590.009
**Figure supplement 1.** PC firing activity in TRPC3 mutants in vitro.
DOI: https://doi.org/10.7554/eLife.45590.008

determine the contribution of TRPC3 to the activity of Z+ and Z– PCs, we performed in vitro electrophysiological recordings on sagittal sections of adult mice of both mutants (*Figure 2B*), taking lobules X and I-III as proxies for Z+ and Z– PC modules, respectively (see *Brochu et al., 1990*; *Sugihara and Quy, 2007*; *Zhou et al., 2014*). In littermate controls, the intrinsic firing rate of Z– PCs is higher than that of Z+ PCs, confirming previous results (*Zhou et al., 2014*). Gain-of-function *TRPC3*$^{Mwk/-}$ mice showed an increase in PC simple spike firing rate selectively in Z– PCs (84.5 ± 36.2 Hz vs. 58.4 ± 19.6 Hz for mutants vs. controls; $t_{19}$ = −2.47, p=0.020), without affecting Z+ PCs (36.7 ± 13.0 Hz vs. 39.7 ± 21.5 Hz for mutants vs. controls $t_{21}$=0.419, p=0.680) (*Figure 2C*). Inversely, ablating TRPC3 from PCs caused a decrease in firing rate in Z– PCs (44.1 ± 15.6 Hz vs. 55.4 ± 21.8 Hz, mut. vs. ctrl, $t_{81}$=2.69, p=0.009), again without affecting Z+ PCs (28.5 ± 9.3 Hz vs. 30.8 ± 11.7 Hz, mut. vs. ctrl, $t_{64}$=0.937, p=0.352) (*Figure 2D*). However, in the absence of TRPC3 the firing rate of Z– PCs does not drop to the levels of Z+ PCs, suggesting that TRPC3 provides a major, but not exclusive, contribution to the difference. We also assessed the regularity of firing activities by measuring the coefficient of variation (CV) and the coefficient of variation of adjacent intervals (CV2) of ISI. Both the CV and CV2 of Z– PCs in lobules I-III declined significantly in *pcp2*$^{Cre}$;*TRPC3*$^{fl/fl}$ mice, while remaining unchanged in *TRPC3*$^{Mwk/-}$ mice; in contrast, in Z+ lobule X, none of these parameters were altered in either *TRPC3*$^{Mwk/-}$ or *pcp2*$^{Cre}$;*TRPC3*$^{fl/fl}$ mice (*Figure 2—figure supplement 1A–B*).

To verify the effect of TRPC3 deletion on other cell physiological properties of PCs, we performed whole-cell patch-clamp recordings in a subset of PCs. Injections of current steps into PCs evoked increasing numbers of action potential, in the presence of blockers for both excitatory and inhibitory synaptic inputs. In line with the cell-attached recordings, in loss-of-function *pcp2*$^{Cre}$; *TRPC3*$^{fl/fl}$ mice, PC intrinsic excitability, quantified by the slope of firing rate versus current injection curve, was significantly reduced in lobules I-III (16.0 ± 1.0 Hz/100 pA vs. 19.2 ± 1.1 Hz/100 pA, mut. vs. ctrl, $t_{32}$=-2.20, p=0.035), but unchanged in lobule X, compared with those of littermate controls (10.4 ± 0.5 Hz/100 pA vs. 11.5 ± 1.1 Hz/100 pA, mut. vs. ctrl, $t_{22}$=-0.95, p=0.354) (*Figure 2E*). Other physiological parameters in terms of holding current, amplitudes, half-widths and after-hyperpolarization amplitudes (AHPs), were not significantly affected in either lobules I-III or lobule X (*Figure 2— figure supplement 1C*).

Together, our in vitro recordings from gain- and loss-of-function mutants indicate that TRPC3 selectively controls the activity in Z– PCs, without affecting other cell intrinsic properties. Thus, at least in vitro, TRPC3 contributes to the difference in intrinsic firing activity between Z+ and Z– PCs, by directly controlling the intrinsic excitability of Z– PCs.

## TRPC3 regulates the activity of simple spikes selectively in Z– PCs in vivo

To examine the role of TRPC3 in the closed loop, intact cerebellar module, we next performed PC recordings in vivo in adult mice during quiet wakefulness (*Figure 3A*). PCs could be identified during extracellular recordings by the presence of complex spikes, while the consistent presence of a pause in simple spikes following each complex spike confirmed that the recording was obtained from a single unit (*De Zeeuw et al., 2011*). PC recording locations in either Z– lobules I-III or Z+ lobule X were confirmed with iontophoretic injections of biotinylated dextran amine (BDA), which could be identified by immunostaining (*Figure 3B*). As we showed before (*Zhou et al., 2014*; *Zhou et al., 2015*), PCs in Z– modules fired simple spikes at a higher rate than those in Z+ modules (*Figure 3D and F*).

In vivo, in the presence of physiological inputs the PCs in Z– lobules I-III of $TRPC3^{Mwk/-}$ mutants showed an increased simple spike firing rate (110.0 ± 22.6 Hz vs. 89.1 ± 15.3 Hz, mut. vs. ctrl, $t_{60}$=-4.58, p<0.001), whereas the Z+ PCs were unaffected (50.6 ± 13.6 Hz vs. 45.3 ± 10.4 Hz, mut. vs. ctrl, $t_{42}$=-1.47, p=0.148). Conversely, Z– PCs in $pcp2^{Cre}$;$TRPC3^{fl/fl}$ mutants featured a decreased simple spike firing rate (74.4 ± 18.6 Hz vs. 88.5 ± 17.4 Hz, mut. vs. ctrl, $t_{54}$=2.88, p=0.006), but again without changes in PCs of the Z+ lobule X (50.2 ± 15.5 Hz vs. 50.0 ± 12.9 Hz, mut. vs. ctrl, $t_{54}$=-0.053, p=0.958), all compared to those of their littermate controls (*Figure 3C–F*). Thus, here too, the ablation of TRPC3 did not decrease simple spike firing rate in the Z– PCs completely to levels of Z+ PCs (74.4 Hz vs. 50.0 Hz in Z– vs. Z+ of $pcp2^{Cre}$;$TRPC3^{fl/fl}$, respectively). Unlike in vitro, PCs in the $pcp2^{Cre}$;$TRPC3^{fl/fl}$ mice showed comparable CV and CV2 to controls for both Z– and Z+ modules (*Figure 3—figure supplement 1C–D*). The CV of simple spike ISI was, however, prominently elevated in both Z– and Z+ modules in $TRPC3^{Mwk/-}$ mutants (*Figure 3—figure supplement 1A*), while the CV2 did not differ (*Figure 3—figure supplement 1B*). It should be noted that PC regularity in vivo is largely determined by external inputs (compare *Figure 2—figure supplement 1* to *Figure 3—figure supplement 1*), which thereby can offset those intrinsic variations induced by the mutation of TRPC3. The irregular firing activity of PCs in $TRPC3^{Mwk/-}$ mutants, at least for Z+ PCs, may be attributed to impaired function or degeneration of UBCs, while the physiological synaptic input in vivo in $pcp2^{Cre}$;$TRPC3^{fl/fl}$ mice could obscure the regularity changes observed in vitro in these mice.

In short, even in vivo, in the presence of all physiological inputs both gain-of-function and loss-of-function mutations of TRPC3 exclusively affects Z– PCs, with the most pronounced, persistent effect being the mutation-selective influence on simple spike firing rate.

## TRPC3-related effects correlate with zebrin expression and are independent of development

Our results so far have identified selective TRPC3-related effects by comparing lobules I-III and X, as proxies for Z– and Z+ modules. Immunohistochemical analysis indicated that the TRPC3 expression differs substantially between these lobules (*Figure 1—figure supplements 1–2* and *Figure 1—video 1*), suggesting that the effects of gain- and loss-of-function mutations could be directly related to protein levels. Alternatively, other differences in molecular machinery could underlie or further enhance this cellular differentiation, for instance through mGluR1b-related effects. As the difference in TRPC3 expression is minimal or absent in the more lateral parts of the cerebellum (*Figure 1—figure supplements 1–2*), recording the activity of adjacent Z– and Z+ PCs there would solve this question (*Wu and Schonewille, 2018*). To this end, we crossed $pcp2^{Cre}$;$TRPC3^{fl/fl}$ mice with EAAT4$^{GFP}$ mice that express GFP in Z+ PCs to generate $pcp2^{Cre}$;$TRPC3^{fl/fl}$;EAAT4$^{GFP/-}$ mice. Using two-photon microscopy, we identified Z+ and Z– modules on the dorsal surface (lobules IV-VI and simplex) of the cerebellum and recorded PC activity (*Figure 4A–B*). Here, the absence of TRPC3 attenuated the firing rate (36.5 ± 23.2 Hz vs. 72.7 ± 26.5 Hz, mut. vs. ctrl, $t_{28}$=3.99, p<0.001) and enhanced the irregularity (e.g. CV: 0.55 ± 0.16 vs. 0.44 ± 0.10, mut. vs. ctrl, $t_{28}$=-2.27, p=0.031) of Z– PCs even more robustly, without an effect on Z+ PCs (rate: 36.6 ± 19.5 Hz vs. 33.0 ± 9.8 Hz, mut. vs. ctrl, $t_{21}$=-0.550, p=0.588; CV: 0.47 ± 0.22 vs. 0.56 ± 0.26, mut. vs. ctrl, $t_{21}$=0.853, p=0.393) (*Figure 4C–D*, *Figure 4—figure supplement 1A–B*, cf *Figure 3F*). The similar firing rates of Z– and Z+ PCs lacking TRPC3 in these targeted recordings supports the possibility that in some areas TRPC3 is solely responsible for the difference and that further differentiation divides the zebrin-based populations into smaller sub-populations (*Armstrong et al., 2001*). It also fits with the finding that other proteins, for example mGluR1b, influence TRPC3 activity and thereby differentially determine the spiking

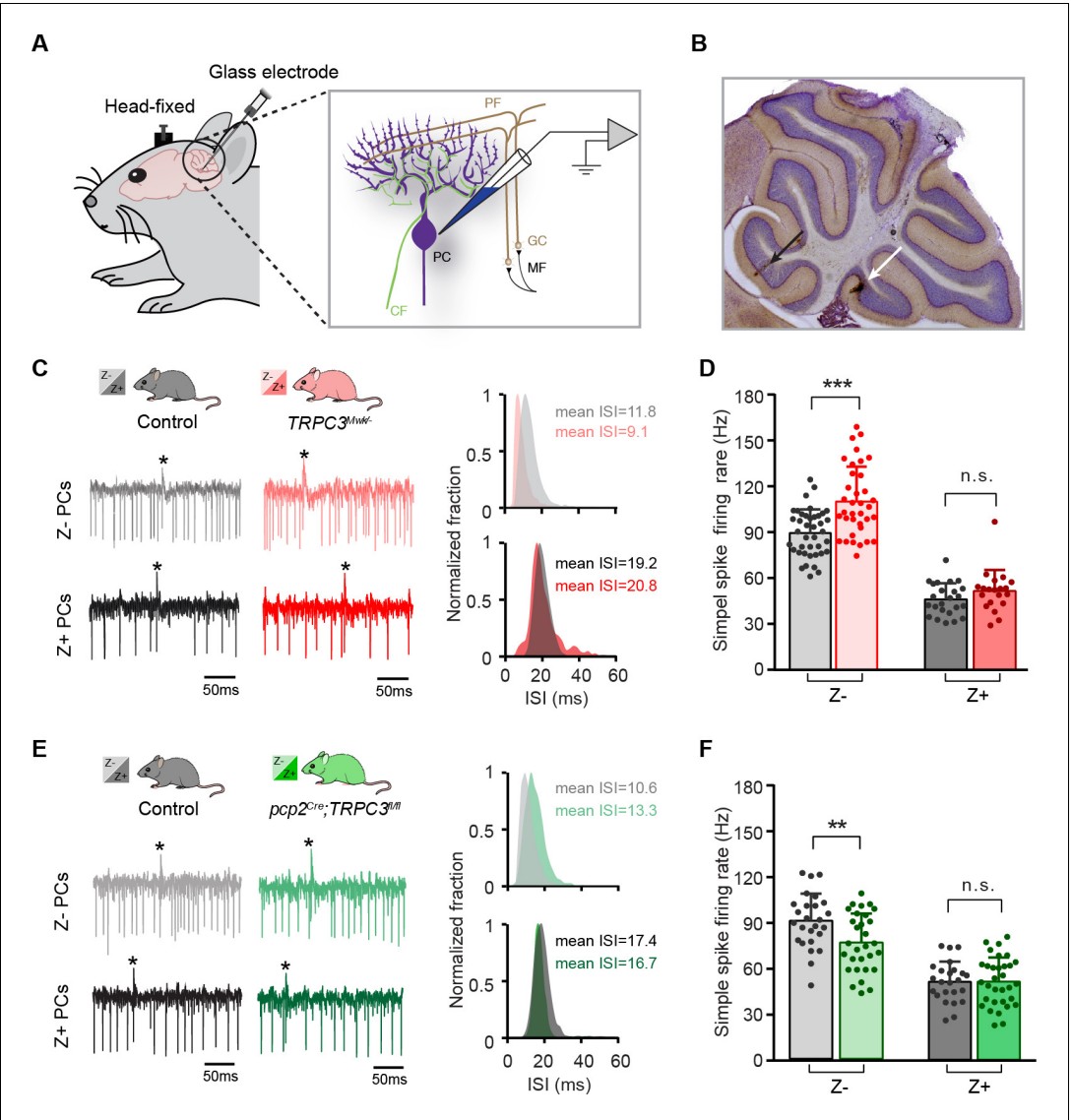

**Figure 3.** TRPC3 contributes to the in vivo simple spike firing rate of Z–, but not Z+ PCs. (A) Schematic illustration of extracellular recording configuration in vivo. PF, parallel fiber; CF, climbing fiber; MF, mossy fiber; GC, granule cell. (B) Representative sagittal cerebellar section with recording sites labeled by BDA injection, in lobule II (black arrow) and X (white arrow). (C) Representative example traces (left) and ISI distributions (right) of a Z– PC (top) and a Z+ PC (bottom) in gain-of-function *TRPC3^{Mwk/-}* mice. (D) PC simple spike firing rate recorded in vivo in *TRPC3^{Mwk/-}* mice compared to control littermates, for the Z–lobules I-III (light-red, n = 36/N = 7 mutants vs. n = 40/N = 6 controls, $t_{60}$ = −4.58, p<0.001) and the Z+ lobule X (dark-red, n = 20/N = 6 mutants vs. n = 24/N = 5 controls, $t_{42}$=-1.47, p=0.148). (E) Representative example traces (left) and ISI distributions (right) in a Z– PC (top) and a Z+ PC (bottom) of loss-of-function *pcp2^{Cre};TRPC3^{fl/fl}* mice. (F) PC simple spike firing rate of *pcp2^{Cre};TRPC3^{fl/fl}* mice compared to controls, for Z– lobules I-III (light-green, n = 30/N = 7 mutants vs. n = 26/N = 8 controls, $t_{54}$=2.88, p=0.006) and in Z+ lobule X (dark-green, n = 32/N = 8 mutants vs. n = 24/N = 6 controls, $t_{54}$ = −0.053, p=0.958). Data are represented as mean ± s.d., for values see Source data, ** means p<0.01 and ***p<0.001.
DOI: https://doi.org/10.7554/eLife.45590.010

The following source data and figure supplement are available for figure 3:

**Source data 1.** Source data for *Figure 3* and supplement.
DOI: https://doi.org/10.7554/eLife.45590.012

**Figure supplement 1.** In vivo extracellular recordings of PC simple spike activity in *TRPC3^{Mwk/-}* and *pcp2^{Cre}; TRPC3^{fl/fl}* mice.
DOI: https://doi.org/10.7554/eLife.45590.011

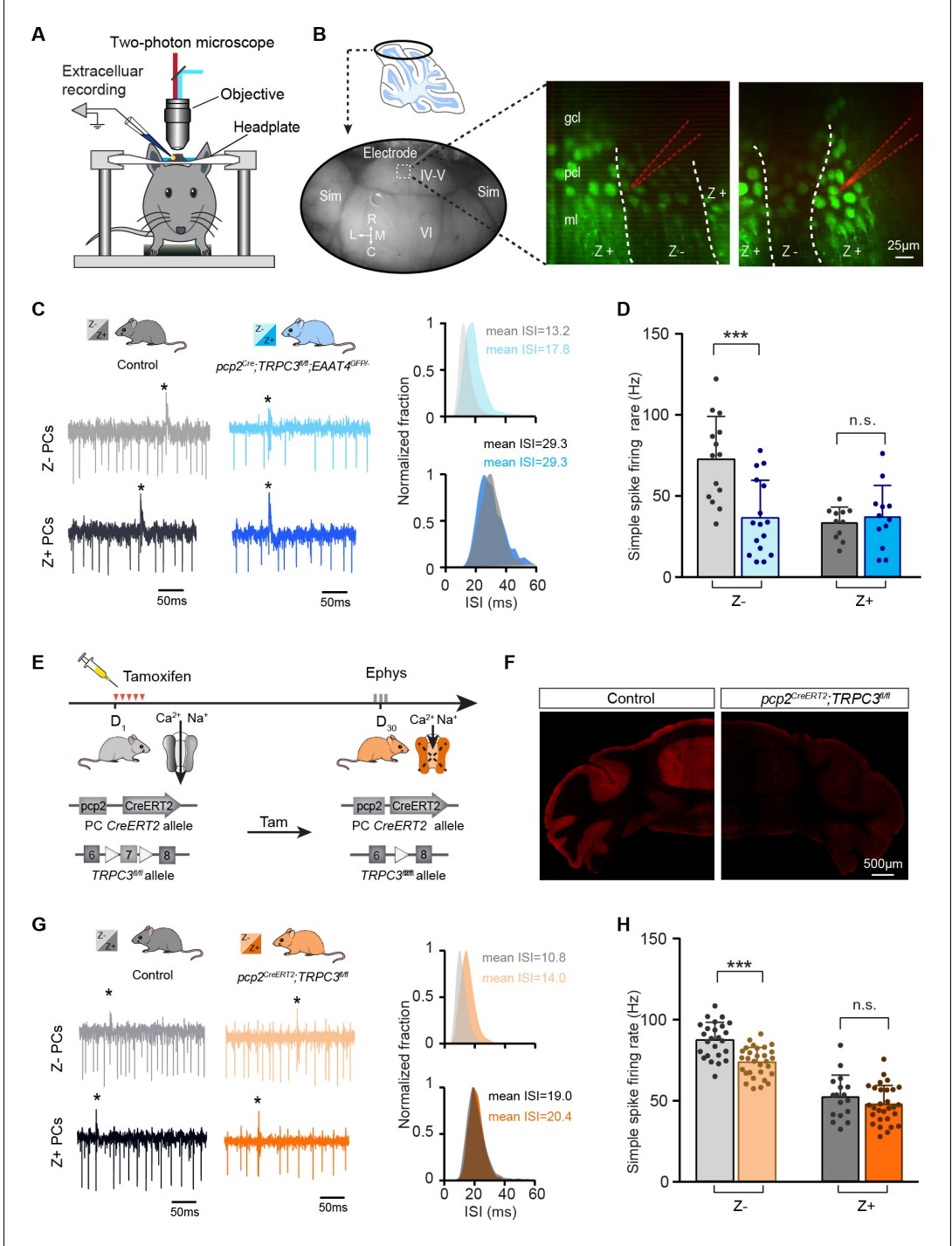

**Figure 4.** The contribution of TRPC3 to firing rate is dependent on zebrin-identity and independent of developmental changes. (**A**) Schematic experimental setup for two-photon imaging-based targeted PC recordings, in vivo. (**B**) Sagittal view of cerebellum (schematic, top) indicating the recording region in the ellipse (bottom). Representative images (right) show the visualization of Z+ bands (dark green) in an awake *pcp2$^{Cre}$;TRPC3$^{fl/fl}$; EAAT4$^{GFP/-}$* mouse, with recording electrodes (red) positioned in Z− (left) and Z+ (right) bands. (**C**) Representative firing traces (left) and ISI distributions (right) in a Z− PC (top) and a Z+ PC (bottom) of loss-of-function *pcp2$^{Cre}$;TRPC3$^{fl/fl}$;EAAT4$^{GFP/-}$* mice (blue) and control littermates (no Cre; gray). (**D**) Average simple spike firing rate of PCs recorded from adjacent modules of *pcp2$^{Cre}$;TRPC3$^{fl/fl}$;EAAT4$^{GFP/-}$* mice and those in controls. Comparison for Z− PCs (light-blue, n = 16/N = 3 mutants vs. n = 14/N = 2 controls, $t_{28}$ = 3.99, p<0.001), and Z+ PCs (dark-blue, n = 12/N = 3 mutants vs. n = 12/N = 2 controls, $t_{21}$ = −0.550, p=0.588). (**E–F**) Intraperitoneal tamoxifen injections for five days (D$_{1-5}$) to trigger TRPC3 gene ablation solely in PCs in adult *pcp2$^{CreERT2}$;TRPC3$^{fl/fl}$* mice. Open triangles indicate *loxP* sites. PC in vivo extracellular activity was recorded four weeks later (D$_{29-31}$) in *pcp2$^{CreERT2}$; TRPC3$^{fl/fl}$* mice (orange). TRPC3 deletion was confirmed post-mortem by confocal imaging following anti-TRPC3 staining (**F**). (**G**) Representative firing

*Figure 4 continued on next page*

Figure 4 continued

traces (left) and ISI distributions (right) in a Z– PC (top) and a Z+ PC (bottom) of $pcp2^{CreERT2}$;$TRPC3^{fl/fl}$ mice. (H) Simple spike firing rate in vivo in $pcp2^{CreERT2}$;$TRPC3^{fl/fl}$ and control mice (no Cre) recorded in lobules I-III (Z–) and lobule X (Z+) PCs. Comparison for Z– PCs (light-orange, n = 30/N = 4 mutants vs. n = 25/N = 4 controls, $t_{53}$ = 5.05, p<0.001), and Z+ PCs (dark-orange, n = 29/N = 4 mutants vs. n = 17/N = 3 controls, $t_{44}$ = 1.21, p=0.234). Sim, simplex lobule; IV-VI, lobules IV-VI, R, rostral, C, caudal; L, lateral, M, medial. Data are represented as mean ± s.d., for values see Source data, *** means p<0.001.

DOI: https://doi.org/10.7554/eLife.45590.013

The following source data and figure supplement are available for figure 4:

**Source data 1.** Source data for *Figure 4* and supplement.

DOI: https://doi.org/10.7554/eLife.45590.015

**Figure supplement 1.** In vivo extracellular recordings of PC simple spike activity in $pcp2^{Cre}$;$TRPC3^{fl/fl}$;$EAAT4^{GFP/-}$ and $pcp2^{CreERT2}$;$TRPC3^{fl/fl}$ mice.

DOI: https://doi.org/10.7554/eLife.45590.014

activity of PCs in areas where TRPC3 expression is more homogeneous. As stated above, the experiments in loss-of-function $pcp2^{Cre}$;$TRPC3^{fl/fl}$ mice suggest that TRPC3 cannot account for the entire difference between Z+ and Z– PCs. As the L7 promotor turns on early in development (postnatal week 1–2; *Barski et al., 2000*), it could be that the ablation of TRPC3 early in development provokes compensatory mechanisms that limit the decrease in simple spike rate in Z– PCs in adult mice. Alternatively, developmental changes in the activity in the olivocerebellar loop could be partially responsible for the lower firing rate. To test the possibility that developmental effects influenced PC activity in the adult mice, we crossed the *loxP*-flanked TRPC3 mice with tamoxifen-dependent $pcp2^{CreERT2}$ to generate $pcp2^{CreERT2}$;$TRPC3^{fl/fl}$ mice (*Figure 4E*). Four weeks after tamoxifen treatment, $pcp2^{CreERT2}$;$TRPC3^{fl/fl}$ mice showed a virtually complete ablation of TRPC3 in PCs (*Figure 4F*). If the absence of TRPC3 early in development drives compensatory mechanisms or contributes to the low simple spike firing rate in adult Z– PCs, we should observe a larger or smaller effect, respectively, in $pcp2^{CreERT2}$;$TRPC3^{fl/fl}$ adult mice after tamoxifen injections (injected after maturation). In vivo recordings revealed that, again, simple spike firing rates were affected in Z– (from lobules I-III, 72.9 ± 9.1 Hz vs. 86.5 ± 10.9 Hz, mut. vs. ctrl, $t_{53}$ = 5.05, p<0.001), but not Z+ PCs (lobule X, rate: 47.0 ± 11.6 Hz vs. 51.6 ± 13.4 Hz, mut. vs. ctrl, $t_{44}$=1.21, p=0.234) of tamoxifen injected adult $pcp2^{CreERT2}$;$TRPC3^{fl/fl}$ mice (*Figure 4G–H* and *Figure 4—figure supplement 1C–D*), in a manner similar to that in $pcp2^{Cre}$;$TRPC3^{fl/fl}$ mice. To verify the efficiency and selectivity of the inducible PC-specific Cre expression line, these mice were also crossed with Cre-dependent tdTomato expressing (Ai14) mice and injected with tamoxifen in the same manner. Confocal images confirm that labeling is virtually exclusively found in PCs (*Figure 4—figure supplement 1E*).

Taken together and combined with $pcp2^{Cre}$;$TRPC3^{fl/fl}$ data, these results indicate that the TRPC3-dependent effects in zebrin-identified PCs are independent of cerebellar development or developmental compensation. Moreover, the larger effect of TRPC3 ablation on Z– PCs in areas where its expression is similar to that in Z+ PCs points towards a further subdivision based on other proteins that might contribute to the simple spike rate in Z– PCs.

## TRPC3 mutations selectively affect the activity in Z– olivocerebellar modules

PCs in the cerebellar cortex, form a closed loop with the cerebellar nuclei neurons they innervate by their axon output and the olivary neurons from which they receive their climbing fiber input (*Ruigrok, 2011*). If TRPC3 contributes to the output of this loop, one could hypothesize that other elements in the loop should be affected by the mutations (*Chaumont et al., 2013*; *Witter et al., 2013*). To test this hypothesis, we examined complex spikes activity in PCs, as the complex spike directly reflects the activity of the climbing fiber and thereby that of the inferior olivary neuron it originates from *Chaumont et al. (2013)*. We identified complex spikes based on their characteristic shape in our in vivo recordings from Z– lobules I-III or Z+ lobule X (*Figure 5A*). Complex spike firing rates were, similar to simple spike rates, higher in Z– than in Z+ PCs (*Figure 5B*), as shown previously (*Zhou et al., 2014*). Chronic manipulations of TRPC3 activity, gain- and loss-of-function, in PCs predominantly affected complex spike firing rate in Z– ($TRPC3^{Mwk/-}$: $t_{68}$=2.68, p=0.009; $pcp2^{Cre}$;$TRPC3^{fl/fl}$: $t_{54}$=2.50, p=0.016; $pcp2^{Cre}$;$TRPC3^{fl/fl}$;$EAAT4^{GFP/-}$: $t_{28}$=3.49, p=0.002), but not Z+ PCs ($TRPC3^{Mwk/-}$: $t_{42}$=1.56, p=0.126; $pcp2^{Cre}$;$TRPC3^{fl/fl}$: $t_{54}$=1.41, p=0.164), except for that in $pcp2^{Cre}$;

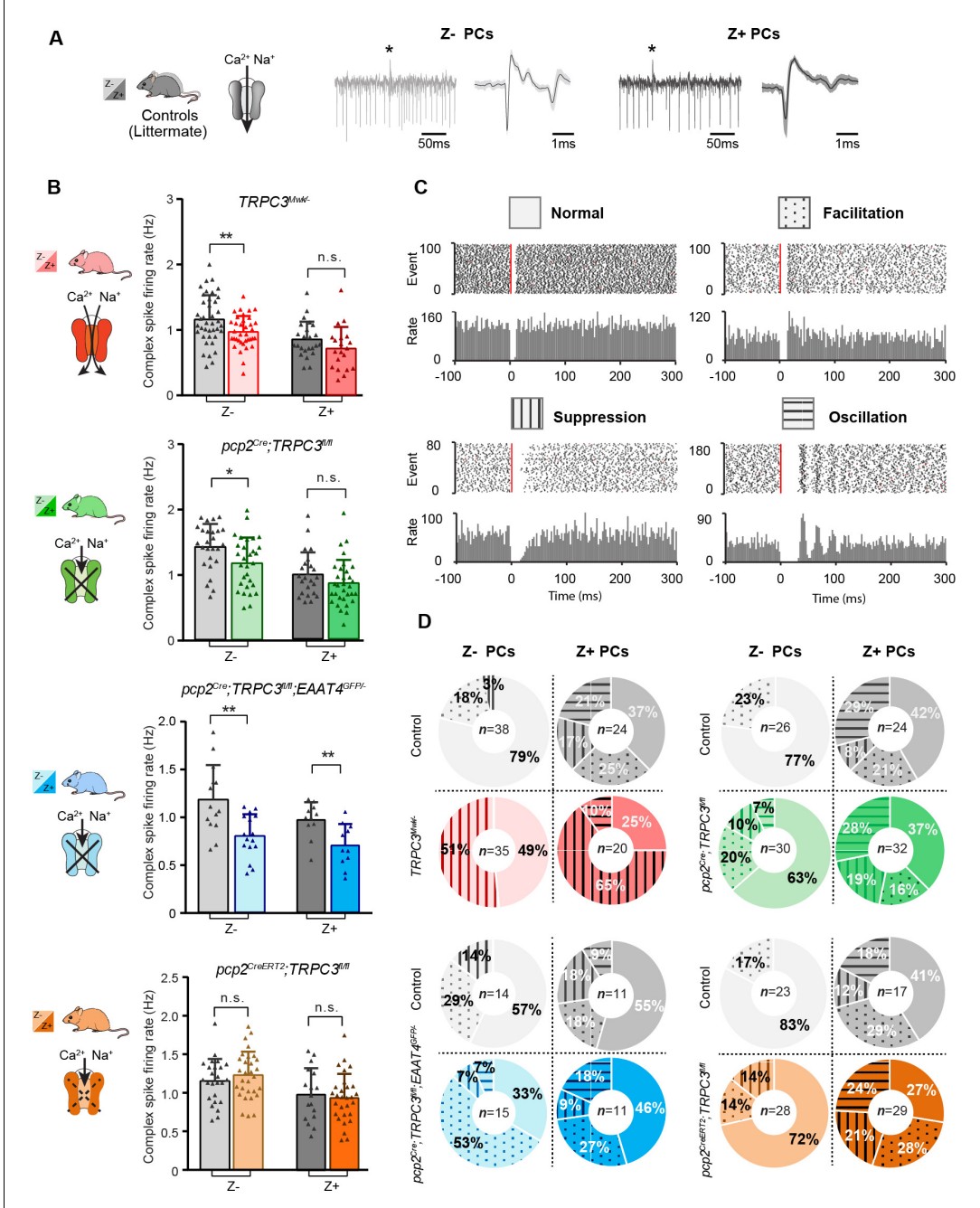

**Figure 5.** Complex spikes and complex spike - simple spike interaction are affected by TRPC3 mutations. (A) Representative PC recording traces and complex spikes shape of Z– (light black) and Z+ (dark black) PCs in the control mice. (B) Top half, comparison of complex spike firing rates in *TRPC3*$^{Mwk/-}$ (red) and *pcp2*$^{Cre}$;*TRPC3*$^{fl/fl}$ (green) mice versus their respective littermate controls for Z– PCs (*TRPC3*$^{Mwk/-}$: $t_{68}$=2.68, p=0.009; *pcp2*$^{Cre}$; *TRPC3*$^{fl/fl}$: $t_{54}$=2.50, p=0.016) and Z+ PCs (*TRPC3*$^{Mwk/-}$: $t_{42}$=1.56, p=0.126; *pcp2*$^{Cre}$;*TRPC3*$^{fl/fl}$: $t_{54}$=1.41, p=0.164). Bottom half, comparison of complex spike firing rates in *pcp2*$^{Cre}$;*TRPC3*$^{fl/fl}$;*EAAT4*$^{GFP/-}$ (blue) and *pcp2*$^{CreERT2}$;*TRPC3*$^{fl/fl}$ (orange) mice versus their respective controls for Z– PCs (*pcp2*$^{Cre}$; *TRPC3*$^{fl/fl}$;*EAAT4*$^{GFP/-}$: $t_{28}$=3.49, p=0.002; *pcp2*$^{CreERT2}$;*TRPC3*$^{fl/fl}$: $t_{53}$=-0.940, p=0.352) and Z+ PCs (*pcp2*$^{Cre}$;*TRPC3*$^{fl/fl}$;*EAAT4*$^{GFP/-}$: $t_{20}$=3.03, p=0.007; *pcp2*$^{CreERT2}$;*TRPC3*$^{fl/fl}$: $t_{44}$=0.448, p=0.656). (C) Raster plots of simple spike activity around the occurrence of each complex spike (−100 to +300 ms). These peri-complex splike time histograms can, based on post-complex spike activity, be divided into one of four types: normal (no change), facilitation, suppression and oscillation. (D) The distribution of post-complex spike response types for Z– and Z+ PCs, in *TRPC3*$^{Mwk/-}$, *pcp2*$^{Cre}$;*TRPC3*$^{fl/fl}$, *pcp2*$^{Cre}$;*TRPC3*$^{fl/fl}$;*EAAT4*$^{GFP/-}$ and *pcp2*$^{CreERT2}$;*TRPC3*$^{fl/fl}$ mice. Data are represented as mean ± s.d., for values see Source data, * means p<0.05 and **p<0.01.

DOI: https://doi.org/10.7554/eLife.45590.016

*Figure 5 continued on next page*

*Figure 5 continued*

The following source data and figure supplement are available for figure 5:

**Source data 1.** Source data for *Figure 5* and supplement.

DOI: https://doi.org/10.7554/eLife.45590.018

**Figure supplement 1.** In vivo extracellular recordings of PC complex spike activity in *TRPC3^{Mwk/-}*, *pcp2^{Cre}*;*TRPC3^{fl/fl}* mice, *pcp2^{Cre}*;*TRPC3^{fl/fl}*;*EAAT4^{GFP/-}* and *pcp2^{CreERT2}*;*TRPC3^{fl/fl}* mice.

DOI: https://doi.org/10.7554/eLife.45590.017

*TRPC3^{fl/fl}*;*EAAT4^{GFP/-}* mice ($t_{20}$ = 3.03, p=0.007) (*Figure 5B*). Intriguingly, acute ablation of TRPC3 in *pcp2^{CreERT2}*;*TRPC3^{fl/fl}* mice did not affect complex spike activity in terms of firing rate, CV, CV2 or pause in simple spikes following climbing fiber activation (CF-pause) in Z– PCs (rate: $t_{53}$=-0.940, p=0.352) (*Figure 5B* bottom panel, *Figure 5—figure supplement 1J–L*). In line with the lower simple spike firing rates in loss-of-function TRPC3 mutants, the CF-pause of *pcp2^{Cre}*;*TRPC3^{fl/fl}* and *pcp2^{Cre}*;*TRPC3^{fl/fl}*;*EAAT4^{GFP/-}* mice were longer, selectively in Z– PCs (*Figure 5—figure supplement 1F and I*). Except for the CV value, other complex spike parameter changes in *TRPC3^{Mwk/-}* mice were not affected (*Figure 5—figure supplement 1A–C*, see also discussion). In *pcp2^{Cre}*;*TRPC3^{fl/fl}* mice, the CV and CV2 of complex spike in both Z– and Z+ PCs do not differ from littermate controls (*Figure 5—figure supplement 1D–E*), however, in *pcp2^{Cre}*;*TRPC3^{fl/fl}*;*EAAT4^{GFP/-}* mice, they were significantly increased for Z– PCs, not Z+ PCs, compared to those of littermate controls (*Figure 5—figure supplement 1G–H*). Together, in vivo experiments indicate that TRPC3 also selectively affects the activity in the inferior olive in that the Z– modules are most prominently affected, and this effect is only present when TRPC3 is deleted early in development.

Complex spikes are known to have a direct influence on simple spike activity (CS-SS) (*Simpson et al., 1996*; *Zhou et al., 2014*). Based on the peri-complex spike time histograms, we could categorize four different types of simple spike responses following the CF-pause (see also *Zhou et al., 2014*), including no change in rate (normal), increased simple spike activity (facilitation), decreased simple spike activity (suppression), and a superimposed oscillatory pattern (oscillation) (*Figure 5C*). Our data confirmed our previous finding that the CS-SS interaction pattern among the Z+ and Z– PCs is different in that the facilitation prevails in the Z– PCs, whereas the suppression and oscillation types occur predominantly in the Z+ PCs (*Figure 5D*), with oscillations seen virtually exclusively in PCs with firing rates < 50 Hz and CV <0.3 (*Zhou et al., 2014*). In addition, we found that manipulation of TRPC3 activity changed the types of CS-SS responses most frequently in Z– PCs (*Figure 5D*). Interestingly, Z– PCs exhibited much more suppression in gain-of-function *TRPC3^{Mwk/-}* mutants and vice versa more facilitation in loss-of-function *pcp2^{Cre}*;*TRPC3^{fl/fl}*;*EAAT4^{GFP/-}* mice, compared to those in their littermate controls (*Figure 5D*), suggesting that Z– PCs partly compensate for the effects of TRPC3 manipulation.

Together, these results indicate that TRPC3 controls not only the activity of PCs, but also that of the inferior olivary neurons, another element in the olivocerebellar loop. Moreover, manipulation of TRPC3 activity alters the interaction between complex spikes and simple spikes.

## Functional heterogeneity of TRPC3 is reflected in differential effects on motor behaviors

The ultimate question is: does cellular heterogeneity of PCs also differentially affect their contribution to specific cerebellar functions? As the *TRPC3^{Mwk/-}* mutation is not cell-specific and affects for instance also UBCs (*Sekerková et al., 2013*), we focused on the behavioral effects in *pcp2^{Cre}*;*TRPC3^{fl/fl}* mice. *pcp2^{Cre}*;*TRPC3^{fl/fl}* mice did not show any overt signs of changes in development or weight, changes in module anatomy or connectivity or signs of any type of locomotion deficit. Before testing specific functions, we first evaluated the consequences of the manipulations of TRPC3 on locomotion, a type of behavior that by nature requires the entire body and as such can be linked to many sub-regions of the cerebellar cortex, ranging from the Z+ vestibular zones to the Z– anterior lobules. We investigated whether these mutant mice showed any obvious deficits in locomotion using the Erasmus Ladder (*Vinueza Veloz et al., 2015*) (*Figure 6—figure supplement 1A*). *pcp2^{Cre}*;*TRPC3^{fl/fl}* mice could not be discriminated from control littermates by the percentage of different types of steps, including lower steps, also known as missteps (*Figure 6—figure supplement 1B–C*).

The apparent discrepancy with earlier evidence in stride width in the global TRPC3 knockout (*Hartmann et al., 2008*) could be due to the different methods or the fact that UBCs, particularly important in the vestibular zone, are also affected in that mouse model (*Sekerková et al., 2013*).

Next, we subjected *pcp2^Cre^;TRPC3^fl/fl^* mice to two specific, but intrinsically distinct types of cerebellum-dependent learning tasks, that is, vestibulo-ocular reflex (VOR) adaptation and eyeblink conditioning. VOR adaptation is the adjustment of the amplitude and/or direction of compensatory eye movements controlled by the vestibulocerebellum (*Figure 6A–C*), which is predominantly Z+ (*Figure 6—figure supplement 2A–B*). Eyeblink conditioning requires the animal to generate a well-timed movement following a previously unrelated sensory input and is linked to more anterior regions that are largely Z– (*Figure 7A* and *Figure 6—figure supplement 2A–B*). Note that the difference in zebrin labeling is pronounced between the two related regions; while the difference in TRPC3 staining is less clear (*Figure 6—figure supplement 2A–B*). Nonetheless, given the electrophysiological changes described above, we hypothesized that altered TRPC3 function should impair Z– linked eyeblink conditioning, whereas VOR adaptation would be unaffected.

Before examining adaptation, we first tested whether the basal eye movement reflexes, the optokinetic reflex driven by visual input (OKR) and the vestibular input-driven VOR (in the dark) and visually-enhanced VOR (VVOR, in the light), were affected. Neither the gain (the ratio of eye movement to stimulus amplitude), nor the phase (timing of the response relative to input), differed significantly between *pcp2^Cre^;TRPC3^fl/fl^* mutants and littermate controls (all p>0.25) (*Figure 6—figure supplement 2C*). Next, using mismatched visual and vestibular stimulation, we tested the ability of mutant mice to adapt their compensatory eye movements. When *pcp2^Cre^;TRPC3^fl/fl^* mice were subjected to both out-of-phase and in-phase training paradigms, we did not observe any significant deficit in the VOR gain increase and VOR gain decrease, respectively (VOR increase, $F = 0.012$, p=0.913; VOR decrease, $F = 0.252$, p=0.621) (*Figure 6D–E*). To evaluate the ability of the mice to perform a long-term, more demanding adaptation, we subjected the mice for four more days, following the gain decrease training, to a training stimulus aimed at reversing the direction of their VOR, referred to as VOR phase reversal (*Figure 6G*). Again, no difference was found between *pcp2^Cre^;TRPC3^fl/fl^* and control littermate mice: neither in the VOR phase over the training (*Figure 6H*), nor in the increased OKR gain following the phase reversal training (VOR phase reversal, $F = 0.006$, p=0.942; OKR increase, $F = 0.922$, p=0.922)(*Figure 6F*, compare to *Figure 6—figure supplement 2C*).

To determine whether the differential activity of TRPC3 ultimately also affects the behavior of the animal, we subjected mice to a task linked to Z– modules, that is delay eyeblink conditioning. Mice were trained using a light pulse with 250 ms duration as the conditioned stimulus (CS) and a puff to the cornea as a short unconditioned stimulus (US) at the end of the CS, which over the period of several days evoked conditioned responses (CR, preventative eyelid closure) in the absence of the US (*Figure 7B*). In contrast to VOR adaptation, the L7-TPRC3^KO^ mice showed significant deficits in eyeblink conditioning during the first week of training (*Figure 7C*). However, when we subjected them to longer periods, they reached similar CR percentages, amplitudes and timing (*Figure 7D* and *Figure 7—figure supplement 1A–B*). The delayed appearance of evoked conditioned responses could neither be explained by a deficit in the ability to close the eyelid, as the timing of the unconditioned response (UR) did not differ between mutant and control mice (*Figure 7—figure supplement 1C*), nor by a lower level of locomotor activity (*Figure 6—figure supplement 1*), which has previously been shown to impair eyeblink conditioning (*Albergaria et al., 2018*).

Thus, although TRPC3 is expressed in both regions underlying the cerebellum-dependent behavioral experiments tested here, TRPC3 activity is selectively required to optimize the cerebellum-dependent learning behavior that is processed in a Z– module (*De Zeeuw and Ten Brinke, 2015*). This indicates that the cellular heterogeneity and consequential differentiation in cellular activity also affects the behavior of the animals.

## Discussion

The cerebellum offers a rich repertoire of electrophysiological properties that allows us to coordinate a wide variety of sensorimotor and cognitive behaviors. We recently uncovered that there are at least two main types of cerebellar modules with different intrinsic profiles (*Zhou et al., 2014*) and plasticity rules (*Wadiche and Jahr, 2005*; *Suvrathan et al., 2016*; *Voges et al., 2017*). This organization is highly preserved throughout phylogeny and characterized by a series of molecular markers

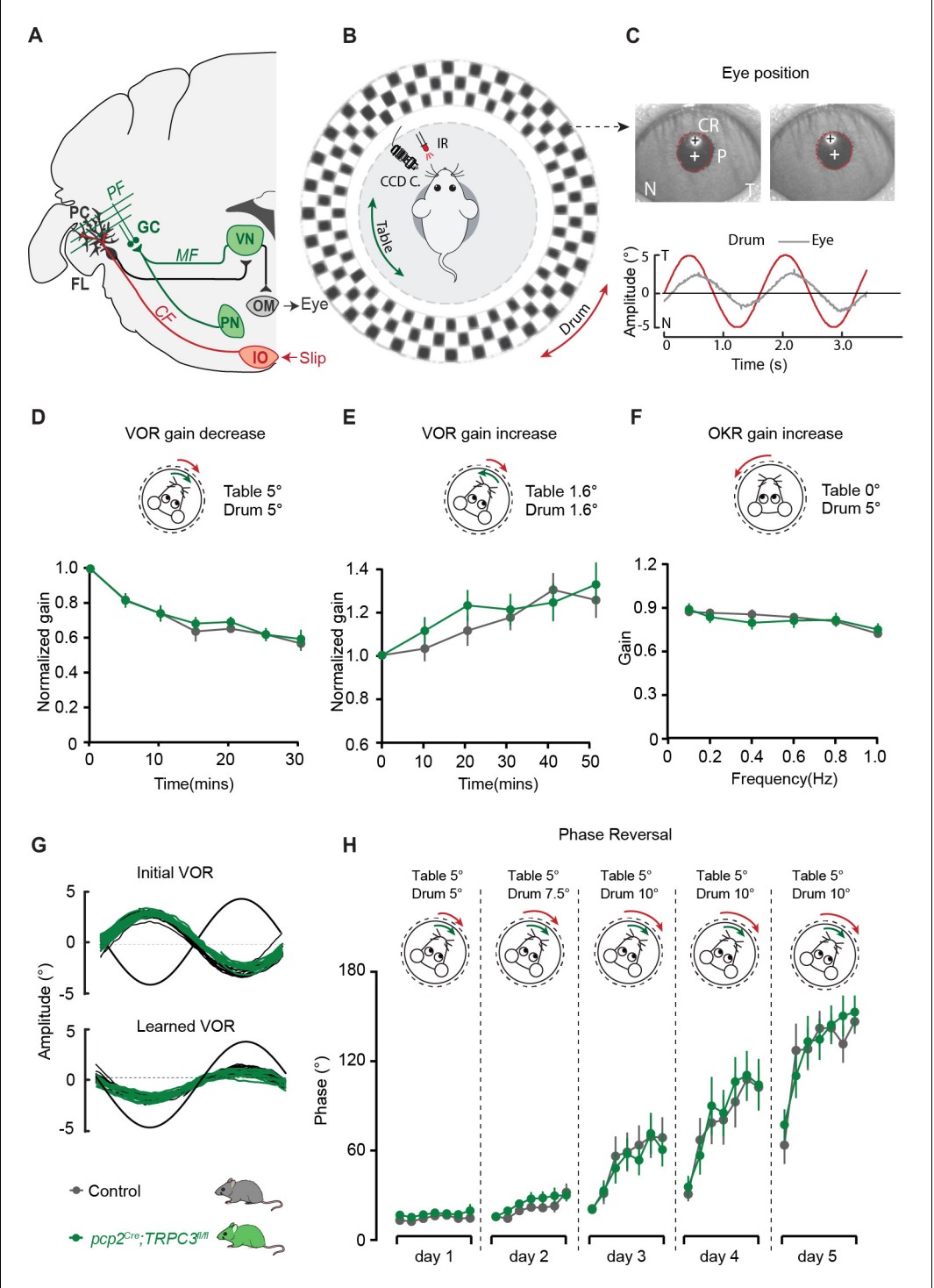

**Figure 6.** PC-specific deletion of TRPC3 does not affect Z+-dependent VOR adaptation. (**A**) Cerebellar circuitry controlling compensatory eye movements and their adaptation. PCs in the flocculus (FL) receive vestibular and visual input via the mossy fiber (MF) - parallel fiber (PF) system (green) and climbing fiber input (CF, red) from the inferior olive (IO), indicating retinal slip. These two inputs converge on PCs, which influence eye movements via the vestibular nuclei (VN) and the oculomotor (OM) neurons. PN, pontine nuclei; GC, granule cell. (**B**) Schematic illustration of eye movement recording setup. Mice are head-fixed in the center of a turntable for vestibular stimulation and surrounded by a random dotted pattern ('drum') for visual stimulation. A CCD camera was used for infrared (IR) video-tracking of the left eye. (**C**) Top, examples of nasal (N) and temporal (T) eye positions. Red

*Figure 6 continued on next page*

*Figure 6 continued*

circles, pupil fit; black cross, corneal reflection (CR); white cross, pupil center. Bottom, example trace of eye position (gray) with drum position (red), during stimulation at an amplitude of 5° and frequency of 0.6 Hz. (D) *pcp2^Cre^;TRPC3^fl/fl^* and control mice were subjected to six 5 min training sessions with mismatched in-phase visual and vestibular stimulation (in light, see insets), aimed at decreasing the VOR gain (probed in the dark before, between and after sessions). (E) Similar, but now mice were trained with out-of-phase stimulation, aimed at increasing VOR gain. (F) Re-recording of OKR gain following the VOR phase reversal training (see **G–H**) to test OKR gain increase (compare to *Figure 6—figure supplement 2C*, left). (G) Multiple-day training using in-phase mismatch stimulation (see inset in **H**) aimed at reversing the direction of the VOR (quantified as a reversal of the phase). Representative eye position recordings of VOR before (top) and after (bottom) training. (H) Results of five days of VOR phase reversal training, probed by recording VOR (in the dark before, between and after sessions) with mice kept in the dark in overnight. Data are represented as mean ± s.e.m., N = 11 mutants versus N = 13 controls, all p>0.05, ANOVA for repeated measurements. See **Source data** for values.

DOI: https://doi.org/10.7554/eLife.45590.019

The following source data and figure supplements are available for figure 6:

**Source data 1.** Source data for *Figure 6* and supplements.

DOI: https://doi.org/10.7554/eLife.45590.022

**Figure supplement 1.** *pcp2^Cre^;TRPC3^fl/fl^* mice show normal Erasmus ladder performance.

DOI: https://doi.org/10.7554/eLife.45590.020

**Figure supplement 2.** Compensatory eye movements and eyeblink conditioning in *pcp2^Cre^;TRPC3^fl/fl^* mice.

DOI: https://doi.org/10.7554/eLife.45590.021

---

such as zebrin that are distributed in a complementary fashion across the cerebellar cortex (*Apps and Hawkes, 2009*; *Marzban and Hawkes, 2011*; *Graham and Wylie, 2012*). Here, we demonstrated that zebrin-negative PCs show a relatively high expression of TRPC3, which has a dominant impact on its electrophysiological features (*Figure 8*). Indeed, gain-of-function and loss-of-function mutations in the gene encoding for TRPC3 selectively affected activity in the zebrin-negative modules and the motor behavior that is controlled by these modules.

Our results confirm previous work indicating that TRPC3 is expressed in all PCs, yet for the first time reveal that its expression is non-uniform and largely complementary to that of well-known marker of cerebellar modules, Zebrin II. Notably, the ablation of TRPC3 decreased the firing rate of Z– PCs to that of Z+ PCs in the superficial areas that were targeted by imaging approaches, but did not completely delete the difference in lobules I-III. Hence, TRPC3 is at least a major contributor to the increased firing rate of Z– PCs, but other factors putatively contribute as well. Although TRPC3 is present in all PCs, loss- and gain-of-function mutations selectively affected Z– PCs, suggesting that other proteins in the pathway leading to TRPC3 activation may be involved. TRPC channels, which are calcium-permeable upon activation by phospholipase C or diacylglycerol, are widely expressed in the brain and critically involved in the development and maintenance of synaptic transmission (*Hartmann et al., 2008*; *Hartmann et al., 2011*; *Becker, 2014*; *Sun et al., 2014*). TRPC1 and TRPC3 are both prominently expressed in the cerebellum, but in PCs TRPC3 is most abundant (*Hartmann et al., 2008*). In addition to its contribution to intrinsic activity, TRPC3 currents also mediate the slow excitatory postsynaptic potential following activation of mGluR1b, which is expressed in a pattern complementary to that of zebrin (*Mateos et al., 2001*; *Hartmann et al., 2011*; *Ohtani et al., 2014*). Our results indicate that TRPC3 can be detected in all PCs in a pattern that is largely, but not completely, complementary to that of Zebrin II, while the effects of TRPC3 ablation are restricted to zebrin-negative PCs. Taken together, this suggests that it is in fact the 'molecular machinery' involving mGluR1b activation combined with TRPC3 expression patterns, that drive the differential TRPC3 function.

In contrast to mGluR1b, mGluR1a is expressed by all PCs (estimated ratio 2:1 to mGluR1b) (*Mateos et al., 2001*). The metabotropic receptor mGluR1a is important for IP3-mediated calcium release, climbing fiber elimination as well as PF-PC LTD (*Ohtani et al., 2014*). Intriguingly, and in line with the concept of modular differentiation, mGluR1-dependent processes are hampered in zebrin-positive PCs by the expression of EAAT4 (*Wadiche and Jahr, 2005*), whereas zebrin-negative PCs selectively express PLCβ4 that works in concert with mGluR1a (*Ohtani et al., 2014*). The differences in expression patterns may enhance the probability of PF-PC LTD in zebrin-negative PCs over

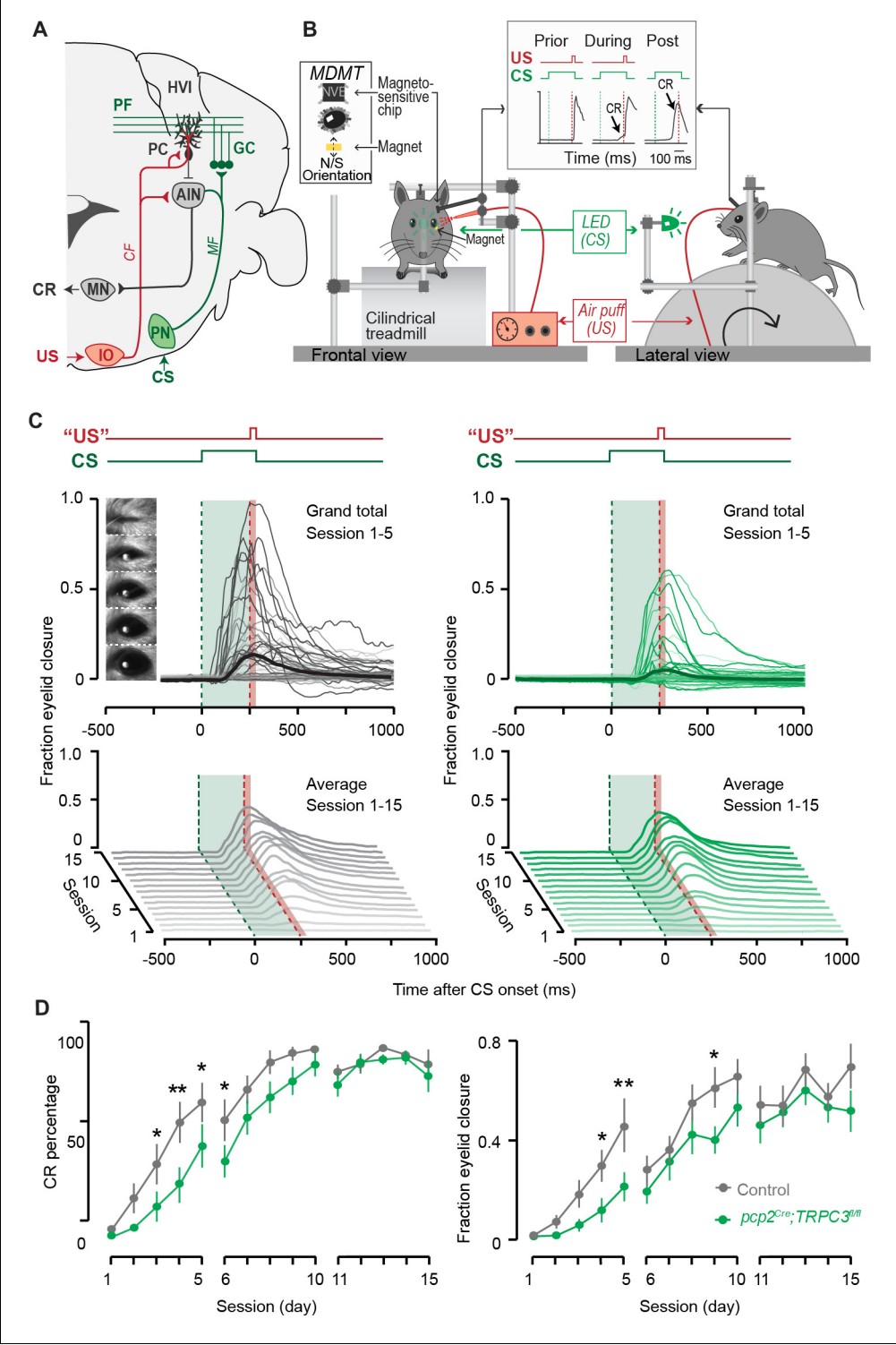

**Figure 7.** Eyeblink conditioning, linked to Z– modules is delayed in *pcp2Cre;TRPC3fl/fl* mice. (**A**) Cerebellar circuitry controlling eyeblink conditioning. PCs in the paravermal region around the primary fissure receive inputs carrying sensory information from for example the pontine nucleus (PN) through the MF-PF pathway and the error signal from the inferior olive (IO) through the climbing fibers (CF). These PCs in turn influence eyelid muscles via the anterior interposed nucleus (AIN) and motor nuclei (MN). (**B**) Schematic illustration of eyeblink conditioning setup. Head fixed mice on a freely moving treadmill, are presented a green LED light (conditional stimulus, CS) followed several hundred milliseconds later by a weak air-puff on the eye (unconditional stimulus, US). As a result of

*Figure 7 continued on next page*

*Figure 7 continued*

repeated CS-US pairings, mice will eventually learn to close their eye in response to the CS, which is called the conditioned response (CR). Eyelid movements were recorded with the magnetic distance measurement technique (MDMT). (C) Comparison of fraction of eyelid closure between controls (left) and *pcp2^Cre^;TRPC3^fl/fl^* mice (right). Top, session averages (thin-lines) per mouse and overall average (thick-lines) for the first 5 days (color intensity increasing from day 1 to 5). Insets: mouse eye video captures show eyelid closure ranging from 0 (fully-open) to 1 (fully-closed). Bottom, waterfall plot of the averaged eyeblink trace during CS-only trials for the 15 daily sessions. (D) The CR percentage and CR amplitude for *pcp2^Cre^;TRPC3^fl/fl^* mice initially have a significantly slower acquisition but eventually reach the same levels as control littermates. Data are represented as mean ± s.e.m., N = 15 mutants versus N = 15 controls, *P* values were all FDR corrected for multiple comparisons, see Source data for values and statistics, * means p<0.05 and **p<0.01.

DOI: https://doi.org/10.7554/eLife.45590.023

The following source data and figure supplement are available for figure 7:

**Source data 1.** Source data for *Figure 7* and supplement.

DOI: https://doi.org/10.7554/eLife.45590.025

**Figure supplement 1.** Eyeblink conditioning in *pcp2^Cre^;TRPC3^fl/fl^* mice.

DOI: https://doi.org/10.7554/eLife.45590.024

that in zebrin-positive PCs, which is supported by experiments performed in P21 mice (*Wadiche and Jahr, 2005*). The consequences of EAAT4 or PLCβ4 deletion on PC physiology have been evaluated in vitro in several studies (*Hashimoto et al., 2001*; *Miyata et al., 2001*; *Wadiche and Jahr, 2005*; *Perkins et al., 2018*), but what the consequences in vivo on circuit physiology and on the behaviors tested here are, is unclear. Our results here demonstrate that changes that occur at the cell physiological level, that is reduced simple spike rate and altered CS-SS interaction, lead to a more complex pattern of changes in the intact system. The additional effects are particularly striking in the *pcp2^Cre^;*

| Zebrin-negative PCs | | | | | | | | | | | | |
|---|---|---|---|---|---|---|---|---|---|---|---|---|
| | In vitro | | | | In vivo | | | | | | | |
| **Mouse line** | Cell-attach | | | Cur-inj | Simple spike | | | Complex spike | | | CF-pause | | |
| | FF | CV | CV2 | FF | FF | CV | CV2 | FF | CV | CV2 | | | |
| **Gain-of-function** *TRPC3^Mwk/-^* | ↑ | — | — | N/A | ↑ | ↑ | — | ↓ | ↑ | — | — | | |
| **Loss-of-function** *pcp2^Cre^;TRPC3^fl/fl^* | ↓ | ↓ | ↓ | ↓ | ↓ | — | — | ↓ | — | — | ↑ | | |
| **Loss-of-function** *pcp2^Cre^;TRPC3^fl/fl^;EAAT4^GFP/-^* | N/A | | | | ↓ | ↑ | ↑ | ↓ | ↑ | ↑ | ↑ | | |
| **loss-of-function** *pcp2^CreERT2^;TRPC3^fl/fl^* | N/A | | | | ↓ | — | — | — | — | — | — | | |

| Zebrin-positive PCs | | | | | | | | | | | | |
|---|---|---|---|---|---|---|---|---|---|---|---|---|
| | In vitro | | | | In vivo | | | | | | | |
| **Mouse line** | Cell-attach | | | Cur-inj | Simple spike | | | Complex spike | | | CF-pause | | |
| | FF | CV | CV2 | FF | FF | CV | CV2 | FF | CV | CV2 | | | |
| **Gain-of-function** *TRPC3^Mwk/-^* | — | — | — | N/A | — | ↑ | — | — | ↑ | — | — | | |
| **Loss-of-function** *pcp2^Cre^;TRPC3^fl/fl^* | — | — | — | — | — | — | — | — | — | — | — | | |
| **Loss-of-function** *pcp2^Cre^;TRPC3^fl/fl^;EAAT4^GFP/-^* | N/A | | | | — | — | — | ↓ | — | — | — | | |
| **loss-of-function** *pcp2^CreERT2^;TRPC3^fl/fl^* | N/A | | | | — | — | — | — | — | — | — | | |

**Figure 8.** Summary of the electrophysiological changes in the gain- and loss-of-function *TRPC3* mutants.

DOI: https://doi.org/10.7554/eLife.45590.026

*TRPC3^{fl/fl}* mice, where the reduced simple spike rate in zebrin-negative PCs leads to a lower complex spike rate. In principle, this could have been a direct olivocerebellar loop effect, as lower simple spike rate results in reduced inhibition of the also inhibitory projection from the cerebellar nuclei to the inferior olive (*Chaumont et al., 2013*; *Witter et al., 2013*). In contrast, *TRPC3^{Mwk/-}* mice exhibit a higher PC simple spike rate, but also a lower complex spike rate. This seems to argue against a direct olivocerebellar loop effect, but the gain-of-function mutation is present in all neurons of this mouse line and thus cell intrinsic processing probably plays a -currently unknown- role herein. The unaltered complex spike rate of *pcp2^{CreERT2}*;*TRPC3^{fl/fl}* mice suggests that there is a developmental component to the relationship between complex spikes and simple spikes (*Badura et al., 2013*; *White and Sillitoe, 2017*).

It should be noted that in some experiments we found that also the Z+ PCs were affected by TRPC3 mutations. In *TRPC3^{Mwk/-}* mice the regularity of simple spikes and complex spikes was higher compared to controls. TRPC3 is expressed in UBCs and inferior olivary neurons (see also *Figure 1* and *Figure 1—figure supplements 1–2*). UBCs are neurons that fire action potential with high regularity (*Ruigrok et al., 2011*) and provide mossy fiber input in lobule X, which -indirectly- drives PC simple spike activity (*Nunzi et al., 2001*), whereas inferior olivary neurons directly control the rate and regularity of complex spikes (*De Zeeuw et al., 2011*). UBCs degenerate early in development in *TRPC3^{Mwk/-}*, while the effects of the mutation on inferior olivary neurons have not been described yet. Thus, the global nature of this mutation could drive changes in other cell types that explain the effects observed in the regularity of simple and complex spikes. In addition, it should be noted that the complex spike rate of Z+ PCs in *pcp2^{Cre}*;*TRPC3^{fl/fl}* mice as observed with the use of 2-photon imaging, was also lower, for which we at current do not have an explanation. In general, in all conditions the complex spikes rate in Z– and Z+ appear to be significantly or trending towards lower values in mutant mice, except for those in inducible mice where the mutation occurred after development. This has two potential implications: 1) TRPC3 mutations, through direct effects on inferior olivary neurons and/or indirect effects through the olivocerebellar loop, have an inhibiting effect on complex spike rates, and 2) these effects are absent when the mutation is induced later in life.

To test the functional consequences of the loss of TRPC3 and the modular specificity of these effects, we tested the impact on behavioral experiments that can be linked to specific modules. Eyeblink conditioning and VOR adaptation are controlled by different modules in the cerebellum and they are distinctly different by nature. Eyeblink conditioning requires a novel, well-timed eyelid movement to a previously unrelated, neutral stimulus, and has been linked to largely or completely zebrin-negative modules in the anterior cerebellum (*Hesslow, 1994*; *Mostofi et al., 2010*). The activity of the putative zebrin-negative PCs in this area is relatively high at rest, in line with their zebrin identity, and a decrease in this high firing rate correlates to the eyeblink response (*Jirenhed et al., 2007*; *Johansson et al., 2014*; *ten Brinke et al., 2015*). Conversely, VOR adaptation adjusts the amplitude of an existing reflex to optimize sensory processing using visual feedback and is controlled by the vestibulocerebellum, the flocculus in particular, which is classically considered to be zebrin-positive (*Sanchez et al., 2002*; *Zhou et al., 2014*; cf *Sugihara and Quy, 2007*; *Fujita et al., 2014*). There are several variations in VOR adaptation aimed at different changes in temporal and/or spatial parameters (see e.g. *Voges et al., 2017*). In unidirectional VOR gain increase, we recently found that the change correlating with the adapted eye movement consisted of a potentiation, an increase, of the -at rest- lower PC firing rate (*Voges et al., 2017*). Although our current study has its main focus on the differential contribution of TRPC3 at the cell and systems physiological level, it is tempting to speculate how the loss of TRPC3 in PCs results in an eyeblink conditioning phenotype without affecting VOR adaptation. The reduction in firing rate of zebrin-negative PCs may directly contribute to the impaired conditioning. The suppression of simple spike firing that generally correlates with the conditioned response (*ten Brinke et al., 2015*) might be occluded by the lower resting rate in *pcp2^{Cre}*;*TRPC3^{fl/fl}* mice. Alternatively, PF-PC LTD could play a role as it is in line with the simple spike suppression and blocking TRPC3 function completely abolishes this form of LTD (*Kim, 2013*). However, genetically ablating PF-PC LTD did not affect the ability to perform eyeblink conditioning successfully (*Schonewille et al., 2011*), arguing against an exclusive role for this form of plasticity. Schreurs and colleagues demonstrated that intrinsic excitability is increased after eyeblink conditioning (*Schreurs et al., 1997*). A third option could be that TRPC3 also affects the adaptive increase of excitability, intrinsic plasticity, which is calcium-activated

potassium channel function dependent (*Ohtsuki et al., 2012*), and thereby delays the expression of a conditioned blink response. All three options would not necessarily affect VOR adaptation and could contribute to the deficits in eyeblink conditioning, but given the relatively mild phenotype, one or two could be sufficient. Future experiments will have to unravel the cellular changes underlying eyeblink conditioning and VOR adaptation and the specific role of TRPC3 in the former.

In this study we aimed to gain insight in the mechanisms that convert molecular heterogeneity into differentiation of cell physiology and function. This mechanistic question goes hand in hand with the more conceptual question: why are there, at least, two different types of PCs? An appealing hypothesis is that zebrin-negative and zebrin-positive bands control two muscles with opposing functions, for example a flexor and an extensor. Trans-synaptic retrograde tracing using rabies virus from antagonist muscles demonstrated that there is no robust division in zebrin-negative and zebrin-positive strips, but that a partial segregation could not be excluded (*Ruigrok et al., 2008*). A second possibility would be that individual muscles are controlled by either only zebrin-negative or zebrin-positive strips, or a combination of both, when needed. In the vestibulocerebellum of the pigeon, each movement direction is controlled by a set of zebrin-negative and zebrin-positive bands (*Graham and Wylie, 2012*). In this configuration each PC within the set, or separately, would then serve a distinct function, for which it is optimized by gene expression patterns. This dissociation of function could entail for example timing versus coordination (*Diedrichsen et al., 2007*) or moving versus holding still (*Shadmehr, 2017*), although none of these distinctions have been linked to specific zebrin-identified modules. Alternatively, it may be the net polarity of the connectivity downstream of the cerebellar nuclei up to the motor neurons or the cerebral cortical neurons that determines the demand(s) of the module(s) involved (*De Zeeuw and Ten Brinke, 2015*). Module-specific driver lines would greatly aid to answer these questions, but are currently not available.

To summarize, our results support the hypothesis that cerebellar modules control distinct behaviors based on cellular heterogeneity, with differential molecular configurations. We present the first evidence for a non-uniform expression pattern of TRPC3 in PCs, complementary to that of zebrin in the vermis but more homogeneous in the hemispheres. Nonetheless, TRPC3 effects are directly coupled to zebrin, a specificity that putatively requires mGluR1b (*Ohtani et al., 2014*), the activator of TRPC3 that is expressed in a pattern perfectly complementary to zebrin (*Mateos et al., 2001*).

Since the discovery of protein expression patterns in the cerebellar cortex (*Brochu et al., 1990*), numerous other proteins with patterned expression have been identified (*Cerminara et al., 2015*). These patterns have been linked to circuit organizations of modules (*Apps and Hawkes, 2009*), to disease and degeneration (*Cerminara et al., 2015*), and more recently to electrophysiological differences (*Wadiche and Jahr, 2005*; *Zhou et al., 2014*). Most expression patterns follow or complement that of zebrin II, but alternative patterns have been observed (*Armstrong et al., 2001*). These patterns commonly further subdivided one of the two populations studied here and thus potentially underlie the remaining variation and differences between lobules (*Apps et al., 2018*). Altogether, this work demonstrates that proper cerebellar function is based on the presence of (at least) two *modi operandi* that have distinct molecular machineries, with TRPC3 as one of the major contributing factors, so as to differentially control sensorimotor integration in downstream circuitries that require control with opposite polarity.

## Materials and methods

**Key resources table**

| Reagent type (species) or resource | Designation | Source or reference | Identifiers | Additional information |
|---|---|---|---|---|
| Species *Mus musculus* | C57BL/6J mice | Charles Rivers | IMSR_JAX:000664 | |
| Species *Mus musculus* | TRPC3[Mwk/-] | *Becker et al., 2009* | MGI:3689326 | F1 of (original) C3H/HeH and C57BL/6J background |
| Species *Mus musculus* | TRPC3[fl/fl] | *Hartmann et al., 2008* | MGI:5451202 | C57BL/6J background |

*Continued on next page*

*Continued*

| Reagent type (species) or resource | Designation | Source or reference | Identifiers | Additional information |
|---|---|---|---|---|
| Species *Mus musculus* | pcp2*Cre* | *Barski et al., 2000* | MGI:2137515 | C57BL/6Jbackground |
| Species *Mus musculus* | pcp2*CreERT2* | The Institut Clinique de la Souris, www.ics-mci.fr | | C57BL/6Jbackground |
| Species *Mus musculus* | EAAT4*GFP/-* | *Gincel et al., 2007* | | C57BL/6Jbackground |
| Species *Mus musculus* | Ai14 | https://www.jax.org/strain/007908 | MGI:3809524 | C57BL/6J background |
| Antibody | Rabbit anti-TRPC3 | Cell Signaling | Cat.#: 77934 | IHC (1:500), WB (1:1000) |
| Antibody | Mouse anti-actin | Millipore | Cat.#: MAB1501 | WB (1:1000) |
| Antibody | Goat anti-Zebrin II/ Aldolase C | Santa Cruz Biotechnology | Cat.#: SC-12065 | IHC (1:500) |
| Antibody | Mouse anti-calbindin | Sigma | Cat.#: C9848 | IHC (1:7000) |
| Antibody | Rabbit anti-GFP | Abcam | Cat. # 290 | IHC (1:1000) |
| Chemical compound, drug | Dextran, Biotin, 3000 MW, Lysine Fixable (BDA-3000) | Thermo Fisher Scientific | D7135 | |
| Software, algorithm | MATLAB v2014a | Mathworks | RRID: SCR_001622 | |
| Software, algorithm | Clampfit 10 | Molecular Devices | RRID: BDSC_14352 | |
| Software, algorithm | Patchmaster software (for in vitro recording analysis) | HEKA Electronics | | |
| Software, algorithm | Spiketrain software (for in vivo recording analysis) | Used under Neurasmus license, currently: kai.voges@nus.edu.sg | | |
| Software, algorithm | Erasmus Ladder 2.0 analysis | Noldus, Wageningen, Netherlands | | |
| Software, algorithm | Compensatory eye movements analysis | https://github.com/MSchonewille/iMove | | |
| Software, algorithm | Eyeblink conditioning analysis | Neurasmus BV, Rotterdam, Netherlands | | |
| Software, algorithm | GraphPad Prism 6 | GraphPad | RRID: SCR_002798 | |
| Software, algorithm | SPSS 20.0 | IBM SPSS | RRID: SCR_002865 | |

## Animals

For all experiments, we used adult male and female mice with a C57Bl/6 background that were, unless stated otherwise, individually housed, had food ab libitum and were on a 12:12 light/dark cycle. In all experiments the experimenters were blind to mouse genotypes. All experiments were approved by the Dutch Ethical Committee for animal experiments and were in accordance with the Institutional Animal Care and Use Committee.

The generation of $TRPC3^{Mwk/-}$ mice has been described previously (*Becker et al., 2009*). Briefly, male BALB/cAnN mice carrying the *Mwk* mutation which was generated in a large-scale ENU mutagenesis program were subjected to cross with normal C3H/HeH females, and the first filial generation ($F_1$) progeny were screened for a variety of defects. The *Mwk* colony was maintained by repeated backcrossing to C3H/HeH. Experimental mice were generated by crossing C3H/HeH mice heterozygous for the *Mwk* mutation with C57Bl/6 mice. Offspring with the *Mwk* mutation on one allele were classified as gain-of-function TRPC3 Moonwalker mutant (referred to as $TRPC3^{Mwk/-}$) and littermate mice lacking the *Mwk* mutation were used as controls. Note that, the $TRPC3^{Mwk/-}$ mutants present evident ataxic phenotype from a very early age, concomitant with progressive degeneration of UBCs and PCs (*Sekerková et al., 2013*).

Mice in which exon 7 of the *Trpc3* gene was flanked by *loxP* sites ($TRPC3^{fl/fl}$ mice) were bred with mice that express Cre under the *Pcp2* promoter ($L7^{Cre/-}$ mice) (*Barski et al., 2000*). The resulting offspring was genotyped using PCR of genomic DNA extracted from tail or toe by standard procedures. The $F_1$ was crossed again with the $TRPC3^{fl/fl}$ mice. Among the second filial generation ($F_2$), mice homozygous for the *loxP* sites and one *Cre* allele were classified as PC-specific TRPC3 knockout ($L7^{Cre/-}$;$TRPC3^{fl/fl}$, here referred to as $pcp2^{Cre}$;$TRPC3^{fl/fl}$) mice and as controls when Cre was absent ($pcp2^{-/-}$;$TRPC3^{fl/fl}$, here 'littermate controls').

$pcp2^{Cre}$;$TRPC3^{fl/fl}$;$EAAT4^{GFP/-}$ mice were generated by crossing $pcp2^{Cre/-}$;$TRPC3^{fl/fl}$ mice with heterozygous $EAAT^{GFP/-}$ mice which express enhanced green fluorescent protein (GFP) under control of *Eaat4* promotor. The $F_2$ offspring those who expressed $TRPC3^{fl/-}$, $pcp2^{Cre/-}$ and $EAAT^{GFP/-}$ were crossed again with the $TRPC3^{fl/fl}$ mice. Among the $F_3$, mice with a homozygous expression of *floxed-TRPC3*, one *Cre* allele and one $EAAT^{GFP}$ allele ($pcp2^{Cre/-}$;$TRPC3^{fl/fl}$;$EAAT4^{GFP/-}$), were used and referred to as $pcp2^{Cre}$;$TRPC3^{fl/fl}$;$EAAT4^{GFP/-}$ mutant mice and as controls when Cre was absent ($pcp2^{-/-}$;$TRPC3^{fl/fl}$;$EAAT4^{GFP/-}$).

Inducible PC-specific TRPC3 knockouts ($pcp2^{CreERT2}$;$TRPC3^{fl/fl}$) were generated by crossbreeding mice carrying the floxed *TRPC3* with mice expressing the tamoxifen-sensitive Cre recombinase *Cre-ERT2* under the control of the *pcp2* promoter (obtained from the Institut Clinique de la Souris, www.ics-mci.fr) (experimental mice: $pcp2^{Cre-ERT2/-}$;$TRPC3^{fl/fl}$). Tamoxifen was dissolved in corn oil to obtain a 20 mg/ml solution, and intraperitoneally injected into all subjects for consecutive 5 days, four weeks prior to electrophysiological recordings. Injections were performed in adults between 12–31 weeks of age. Efficiency and selectivity of the $pcp2^{Cre-ERT2/-}$ line was verified by crossing mice with Cre-dependent tdTomato expressing (Ai14) mice, injection offspring carrying both alleles with Tamoxifen in a similar manner and analyzing the resulting tdTomato expression. Experimental cohorts were always injected at the same time. Mice without $pcp2^{Cre-ERT2}$ expression were used as controls in this study (experimental mice: $pcp2^{CreERT2}$;$TRPC3^{fl/fl}$).

## Immunohistochemistry

Anesthetized mice were perfused with 4% paraformaldehyde in 0.12M phosphate buffer (PB). Brains were taken out and post-fixed for 1 hr in 4% PFA at room temperature, then transferred in 10% sucrose overnight at 4°C. The next day, the solution was changed for 30% sucrose and left overnight at 4°C. Non-embedded brains were sectioned either sagittally or transversally at 40 μm thickness with freezing microtome. Free-floating sections were rinsed with 0.1M PB and incubated 2 hr in 10 mM sodium citrate at 80°C for 2 hr, for antigen retrieval. For immuno-fluorescence, sections were rinsed with 0.1M PB, followed by 30 min in Phosphate Buffered saline (PBS). Sections were incubated 90 min at room temperature in a solution of PBS/0.5%Triton-X100/10% normal horse serum to block nonspecific protein-binding sites, and incubated 48 hr at 4°C in a solution of PBS/0.4% Triton-X100/2% normal horse serum, with primary antibodies as follows: Aldolase C (1:500, goat polyclonal, SC-12065), Calbindin (1:7000, mouse monoclonal, Sigma, #C9848), and TRPC3 (1:500, rabbit polyclonal, Cell Signaling, #77934). The TRPC3 antibody was validated by comparing results to previous immunostainings and in situ hybridization data (Allen Brain Atlas) and by the selective absence of staining in PCs of PC-specific knockout mice (*Figure 1—figure supplement 3E–F*). After rinsing in PBS, sections were incubated 2 hr at room temperature in PBS/0.4% Triton-X100/2% normal horse serum solution with secondary antibodies coupled with Alexa488, Cy3 or Cy5 (Jackson ImmunoResearch), at a concentration of 1:200. Sections were mounted on coverslip in chrome alum (gelatin/chromate) and covered with Mowiol (Polysciences Inc). For Light Microscopy section were pretreated for endogenous peroxidase activity blocking with 3%$H_2O_2$ in PBS, then rinsed for 30 min in

PBS, incubated 90 min in a solution of PBS/0.5%Triton-X100/10% normal horse serum to block non-specific protein-binding sites, followed by the primary antibody incubation as described before. After 48 hr, sections were rinsed in PBS and incubated 2 hr at room temperature in PBS/0.4% Triton-X100/10% normal horse serum solution with HRP coupled secondary antibodies (Jackson ImmunoResearch), at a concentration of 1:200. Sections were rinsed with 0.1M PB and incubated in diamino-benzidine (DAB, 75 mg/100 ml) for 10 min. Sections were mounted on glasses in chrome alum (gelatin/chromate), dried with successive Ethanol steps, incubated in Xylene and covered with Permount mounting medium (Fisher Chemical). Images were acquired with an upright LSM 700 confocal microscope (Zeiss) for fluorescent microscopy, and Nanozoomer (Hamamatsu) for light microscopy. Fluorescence intensity along the regions of interest were assessed using the 'plot profile' function of Image J.

## iDISCO and light sheet imaging

This protocol has been adapted from a previous study (Renier et al., 2014). After normal perfusion and post-fixation, brains were washed successively in PBS (1.5 hr), 20% methanol/$H_2O$ (1 hr), 50% methanol/$H_2O$ (1 hr), 80% methanol/$H_2O$ (1 hr), and 100% methanol (1 hr) twice. To increase clearance, samples were treated with a solution of dichloromethane (DCM) and 100% methanol (2:1) for another hour. Brains were then bleached with 5% $H_2O_2$ in 90% methanol (ice cold) at 4°C overnight. After bleaching, samples successively washed in 80% methanol/$H_2O$, 50% methanol/$H_2O$, 40% methanol/PBS, and 20% methanol/PBS, for 1 hr each, and finally in PBS/0.2% Triton X-100 for 1 hr twice. After rehydration, samples were pre-treated in a solution of PBS/0.2% Triton X-100/20% DMSO/0.3 M glycine at 37°C for 36 hr, then blocked in a mixture of PBS/0.2% Triton X-100/10% DMSO/6% donkey serum at 37°C for 48 hr. Brains were incubated in primary antibody in PTwH solution (PBS/0.2% Tween-20/5% DMSO/3% donkey serum with 10 mg/ml heparin) for 7 days at 37°C with primary antibody: TRPC3 rabbit polyclonal, 1:500 (Cell Signaling, #77934). Amphotericin was added once every two days at 1 μg/ml to avoid bacterial growth. Samples were then washed in 24 hr in PTwH for six times (1 hr for each, after the fourth wash, leave it at room temperature overnight), followed by the second round of 7 day incubation with primary antibody. Brains were then washed in PTwH, 6 washes in 24 hr, as described before, then incubated in secondary antibody in PTwH/3% donkey serum at 37°C for 7 days with secondary anti-Rabbit Cy3 (Jackson ImmunoResearch) at 1:750. Brains were then washed in PTwH, 6 washes in 24 hr, again, followed by successive washes in 20% methanol/$H_2O$, 40% methanol/$H_2O$, 60% methanol/$H_2O$, 80% methanol/$H_2O$, and 100% methanol twice, for 1 hr each, and finally incubation overnight in a solution of DCM and100% methanol. For tissue clearing, brains were incubated 20 mins in DCM, twice, and conserved in Benzyl ether at room temperature.

Ready samples were imaged in horizontal orientation with an UltraMicroscope II (LaVision BioTec) light sheet microscope equipped with Imspector (version 5.0285.0) software (LaVision BioTec). Images were taken with a Neo sCMOS camera (Andor) (2560 × 2160 pixels. Pixel size: 6.5 × 6.5 μm2). Samples were scanned with double-sided illumination, a sheet NA of 0.148348 (resuls in a 5 μm thick sheet) and a step-size of 2.5 μm using the horizontal focusing light sheet scanning method with the optimal amount of steps and using the contrast blending algorithm. The effective magnification for all images was 1.36x (zoombody*objective + dipping lens = 0.63x*2.152x). Following laser filter combinations were used: Coherent OBIS 488–50 LX Laser with 525/50 nm filter, Coherent OBIS 561–100 LS Laser with 615/40 filter, Coherent OBIS 647–120 LX with 676/29 filter.

## Western blot and fractionation

Cerebellar tissue from $pcp2^{Cre}$;$TRPC3^{fl/fl}$ and control mice was dissected and immediately frozen in liquid nitrogen. Samples were homogenized with a Dounce homogenizer in lysis buffer containing 50 mM Tris-HCl pH 8, 150 mM NaCl, 1% Triton X-100, 0.5% sodium deoxycholate, 0.1% SDS and protease inhibitor cocktail. Protein concentrations were measured using Pierce BCA protein assay kit (Thermo Fisher). Samples were denatured and proteins were separated by SDS-PAGE in Criterion TGX Stain-Free Gels (Bio-Rad), and transferred onto nitrocellulose membranes with the Trans-Blot Turbo Blotting System (Bio-Rad). Membranes were blocked with 5% BSA (Sigma-Aldrich) in TBST (20 mM Tris-HCl pH7.5, 150 mM NaCl and 0.1%, Tween20) for 1 hr and probed with the following primary antibodies: rabbit anti-TRPC3 (Cell Signaling Technology, #77934; 1:1000) and mouse anti-

actin (Millipore, MAB1501; 1:1000). Secondary antibodies used were IRDye 800CW Donkey anti-Rabbit IgG (LI-COR Biosciences, Cat # 925–32213; 1:20000) and IRDye 680RD Donkey anti-Mouse IgG (LI-COR Biosciences, Cat # 925–68072; 1:20000). Membranes were scanned by Odyssey Imager (LI-COR Biosciences) and quantified using Image Studio Lite (LI-COR Biosciences). For quantification, densitometry of protein bands of interest was normalized to that of actin.

For fractionation experiments, cerebellar tissues from C57/BL6 were collected and the synaptosomes were isolated using Syn-PER Synaptic Protein Extraction Reagent (ThermoScientific, #87793) according to the manufacturer's instructions.

## In vivo extracellular recordings and analysis

We performed in vivo extracellular recordings in adult $TRPC3^{Mwk/-}$ (aged 15–47 weeks, ages roughly matched), $pcp2^{Cre};TRPC3^{fl/fl}$ (aged 22–43 weeks, ages roughly matched), $pcp2^{CreERT2};TRPC3^{fl/fl}$ (aged 17–28 weeks) mice, respectively, as previously described (*Zhou et al., 2014*). Briefly, an immobilizing pedestal consisting of a brass holder with a neodymium magnet (4 × 4×2 mm) was fixed on the skull, overlying the frontal and parietal bones, and then a craniotomy (Ø3 mm) was made in the interparietal or occipital bone under general anesthesia with isoflurane/$O_2$ (4% induction, 1.5–2% maintenance). After over 24 hr of recovery, mice were head-fixed and body restrained for recordings. PCs were recorded from vermal lobules I-III and X, using a glass pipette (OD 1.5 mm, ID 0.86 mm, borosilicate, Sutter Instruments, USA; 1–2 µm tips, 4–8 MΩ) with a downward pitch angle of 40˚ and 65˚ respectively. The pipettes were filled with 2 M NaCl-solution and mounted on a digital 3-axis drive (SM-5, Luigs Neumann, Germany). After recording, BDA was iontophoretically injected to confirm that the recordings were from Lobules I-III or X. PCs were identified by the presence of simple and complex spikes, and determined to be from a single unit by confirming that each complex spike was followed by a climbing fiber pause. All in vivo recordings were analyzed offline using Spiketrain (used under Neurasmus license, currently: kai.voges@nus.edu.sg), running under MatLab (Mathworks, MA, USA). For each cell, the firing rate, CV and mean CV2 were determined for simple and complex spikes, as well as the climbing fiber pause. The CV is calculated by dividing the standard deviation, SD, by the mean of ISIs, whereas CV2 is calculated as $2\times|ISI_{n+1}-ISI_n| / (ISI_{n+1}+ISI_n)$. Both are measures for the regularity of the firing, with CV reflecting that of the entire recording and mean CV2 that of adjacent intervals, making the latter a measure of regularity on small timescales. The climbing fiber pause is determined as the duration between a complex spike and the fist following simple spike. To extend this analysis, we also plotted histograms of simple spike activity time locked on the complex spike, and labeled the shape of this time histogram as normal, facilitation, suppression, or oscillation.

## In vivo two-photon-targeted electrophysiology

Details on targeted electrophysiological recordings in vivo in the mouse cerebellum were described previously (*Wu and Schonewille, 2018*). PCs in lobules IV-VI were recorded in adult $pcp2^{Cre};TRPC3^{fl/fl};EAAT4^{GFP/-}$ mice (aged 14–49 weeks, ages roughly matched) under two-photon microscope guidance. A custom-made head plate was fixed to the cleaned skull of each animal, under isoflurane anesthesia, with dental adhesive (Optibond; Kerr Corporation, West Collins, USA) and secured with dental acrylic. A craniotomy was made above the cerebellum, exposing lobules IV-VI. The craniotomy was sealed with biocompatible silicone (Kwik-Cast; World Precision Instruments) and the animal was allowed to recover from surgery before recording. The silicone seal was removed prior to recording. To keep the brain surface moist, Ringer solution containing (in mM): NaCl 135, KCl 5.4, $MgCl_2$ 1, $CaCl_2$ 1.8, HEPES 5 (pH 7.2 with NaOH; Merck, Darmstadt, Germany) was applied. Glass micropipettes with tip size of ~1 µm (resistance: 6–9 MΩ) were advanced from the dorsal surface under a 25˚ angle into the cerebellum, allowing concurrent two-photon imaging with a long working distance objective (LUMPlanFl/IR 40×/0.8; Olympus) on a custom-built two-photon microscope. Pipettes were filled with the same Ringer solution with an additional 40 µM AlexaFluor 594 hydrazide (Sigma-Aldrich, Steinheim, Germany) for visualization. GFP and AlexaFluor 594 were simultaneously excited by a MaiTai laser (Spectra Physics Lasers, Mountain View, CA, USA) operated at 860 nm. Green (GFP) and red (AlexaFluor 594) fluorescence were separated by a dichroic mirror at 560 nm and emission filters centered at 510 nm (Brightline Fluorescence Filter 510/84; Semrock) and 630 nm (D630/60; Chroma), respectively. The brain surface was stabilized with agarose (2% in

Ringer; Sigma–Aldrich) and pipette pressure was initially kept at 3 kPa while entering the brain tissue. It was then removed for cell approach and the actual recording. Extracellular potentials were acquired with a MultiClamp 700A amplifier (Molecular Devices, Sunnyvale, CA, USA) in current-clamp mode. Signals were low-pass filtered at 10 kHz (four-pole Bessel filter) and digitized at 25 kHz (Digidata 1322A). Data were recorded with pCLAMP 9.2 (Molecular Devices). Z+ and Z− cells were identified by comparing the relative intensity of GFP fluorescence. Whenever possible, cells of both types were recording alternatingly between adjacent bands Purkinje neurons with high and low GFP fluorescence.

## In vitro electrophysiology and analysis

We performed in vitro electrophysiological recordings on $TRPC3^{Mwk/-}$ (aged 9–21 weeks, ages roughly matched) and $pcp2^{Cre};TRPC3^{fl/fl}$ (aged 20–60 weeks, ages roughly matched). As described previously (*Peter et al., 2016*), acute sagittal slices (250 μm thick) were prepared from the cerebellar vermis and put into ice-cold slicing medium which contained (in mM) 240 sucrose, 2.5 KCl, 1.25 $Na_2HPO_4$, 2 $MgSO_4$, 1 $CaCl_2$, 26 $NaHCO_3$ and 10 D-Glucose, carbogenated continuously with 95% $O_2$ and 5% $CO_2$. After cutting using a vibrotome (VT1200S, Leica), slices were incubated in artificial cerebrospinal fluid (ACSF) containing (in mM): 124 NaCl, 5 KCl, 1.25 $Na_2HPO_4$, 2 $MgSO_4$, 2 $CaCl_2$, 26 $NaHCO_3$ and 15 D-Glucose, equilibrated with 95% $O_2$ and 5% $CO_2$ at 33.0 ± 1.0°C for 30 min, and then at room temperature. NBQX (10 μM), DL-AP5 (50 μM), and picrotoxin (100 μM) were bath-applied to block AMPA-, NMDA-, and GABA subtype A (GABA$_A$)-receptors, respectively. PCs were identified using visual guidance by DIC video microscopy and water-immersion 40X objective (Axioskop 2 FS plus; Carl Zeiss, Jena, Germany). Recording electrodes (3–5 MΩ, 1.65 mm outside diameter and 1.11 mm interior diameter (World Precision Instruments, Sarasota, FL, USA) were prepared using a P-97 micropipette puller (Sutter Instruments, Novato, CA, USA), and filled with ACSF for cell-attached recordings, or with an intracellular solution containing (in mM): 120 K-Gluconate, 9 KCl, 10 KOH, 4 NaCl, 10 HEPES, 28.5 Sucrose, 4 $Na_2ATP$, 0.4 $Na_3GTP$ (pH 7.25–7.35 with an osmolality of 295) for whole-cell recordings. We measured spontaneous firing activity of PCs in cell-attached mode (0 pA injection) and intrinsic excitability in whole-cell current-clamp mode by injection of brief (1 s) depolarizing current pulses (ranging from −100 to 1100 pA with 100 pA increments) from a membrane holding potential of –65 mV at 33.0 ± 1.0°C. The spike count of evoked action potential was taken as a measure of excitability. AP properties including peak amplitude, AHP and half-width were evaluated using the first action potential generated by each PC. AHP indicates the amplitude of undershoot relative to the resting membrane potential. Half-width indicates the width of the signal at 50% of the maximum amplitude. PCs that required > −800 pA to maintain the holding potential at −65 mV or fired action potentials at this holding potential were discarded. The average spiking rate measured over the entire current pulse was used to construct current-frequency plots. For whole-cell Recordings, cells were excluded if the series (Rs) or input resistances (Ri) changed by >15% during the experiment, which was determined using a hyperpolarizing voltage step relative to the −65 mV holding potential. All electrophysiological recordings were acquired in lobules I-III and lobule X of the vermal cerebellum using EPC9 and EPC10-USB amplifiers (HEKA Electronics, Lambrecht, Germany) and Patchmaster software (HEKA Electronics). Data were analyzed afterwards using Clampfit (Molecular Devices).

## Compensatory eye movement recordings

We subjected alert $pcp2^{Cre};TRPC3^{fl/fl}$ mice (aged 12–39 weeks, ages roughly matched) to compensatory eye movement recordings which were described in detail previously (*Schonewille et al., 2010*). In short, mice were equipped with a pedestal under general anesthesia with isoflurane/$O_2$. After a 2–3 days of recovery, mice were head-fixed with the body loosely restrained in a custom-made restrainer and placed in the center of a turntable (diameter: 63 cm) in the experimental set-up. A round screen (diameter 60 cm) with a random dotted pattern ('drum') surrounded the mouse during the experiment. Compensatory eye movements were induced by sinusoidal rotation of the drum in light (OKR), rotation of the table in the dark (VOR) or the rotation of the table in the light (visually enhanced VOR, VVOR) with an amplitude of 5° at 0.1–1 Hz. Motor performance in response to these stimulations was evaluated by calculating the gain (eye velocity/stimulus velocity) and phase (eye to stimulus in degrees) of the response. Motor learning was studied by subjecting mice to mismatched

visual and vestibular input. Rotating the drum (visual) and table (vestibular) simultaneously, in phase at 0.6 Hz (both with an amplitude of 5°, 5 × 10 min) in the light will induce an increase of the gain of the VOR (in the dark). Subsequently, VOR Phase reversal was tested by continuing the next days (day 2–5, keeping mice in the dark in between experiments) with in phase stimulation, but now with drum amplitudes of 7.5° (days 2) and 10° (days 3, 4, and 5), while the amplitude of the turntable remained 5°. This resulted, over days of training, in the reversal of the VOR direction, from a normal compensatory rightward eye movement (in the dark), when the head turns left, to a reversed response with a leftward eye movement, when the head moves left. At the end of the VOR phase reversal training the OKR was probed again and compared to the OKR before training, to examine OKR gain increase. VOR gain increase was evoked by subjecting mice to out of phase drum and table stimulation at 1.0 Hz (both with an amplitude of 1.6°). A CCD camera was fixed to the turntable in order to monitor the eyes of the mice. Eye movements were recorded with eye-tracking software (ETL-200, ISCAN systems, Burlington, NA, USA). For eye illumination during the experiments, two infrared emitters (output 600 mW, dispersion angle 7°, peak wavelength 880 nm) were fixed to the table and a third emitter, which produced the tracked corneal reflection, was mounted to the camera and aligned horizontally with the optical axis of the camera. Eye movements were calibrated by moving the camera left-right (peak-to-peak 20°) during periods that the eye did not move (*Stahl, 2004*). Gain and phase values of eye movements were calculated using custom-made Matlab routines (MathWorks, https://github.com/MSchonewille/iMove) (*Schonewille, 2019*).

## Eyeblink conditioning

For all procedures on eyeblink conditioning we refer to the study done previously (*Boele et al., 2018*). $pcp2^{Cre};TRPC3^{fl/fl}$ mice, aged 16–25 weeks (ages roughly matched), were anesthetized with an isoflurane/oxygen mixture and surgically placed a so-called pedestal on the skull. After a 2–3 days' recovery, mice were head-fixed and suspended over a foam cylindrical treadmill on which they were allowed to walk freely. Before each session starting, a minuscule magnet (1.5 × 0.7×0.5 mm) was placed on the left lower eyelid with superglue (cyanoacrylate) and an NVE GMR magnetometer was positioned above the left upper eyelid. With this magnetic distance measurement technique (MDMT), we measured the exact positions of each individual mouse eyelid by analyzing the range from optimal closure to complete aperture. The CS was a green LED light (CS duration 280 ms, LED diameter 5 mm) placed 10 cm in front of the mouse's head. The US consisted of a weak air-puff applied to the eye (30 psi, 30 ms duration), which was controlled by an API MPPI-3 pressure injector, and delivered via a 27.5-gauge needle that was perpendicularly positioned at 0.5–1 cm from the center of the left cornea. The training consisted of 3 daily habituation sessions, one baseline measurement, 3 blocks of 5 daily acquisition sessions (each block was separated by 2 days of rest). During the habituation sessions, mice were placed in the setup for 30–45 min, during which the air puff needle (for US delivery) and green LED (for CS delivery) were positioned properly but no stimuli were presented. On the day of acquisition session 1, each animal first received 20 CS-only trials as a baseline measure, to establish that the CS did not elicit any reflexive eyelid closure. During each daily acquisition session, every animal received in total 200 paired CS-US trials, 20 US only trials, and 20 CS only trials. These trials were presented in 20 blocks, each block consisted of 1 US only trial, 10 paired CS-US trials, and 1 CS only trial. Trials within the block were randomly distributed, but the CS only trial was always preceded by at least two paired CS-US trials. The interval between the onset of CS and that of US was set at 250 ms. All experiments were performed at approximately the same time of day by the same experimenter. Individual eyeblink traces were analyzed with Blink 2.0 software (Neurasmus, www.neurasmus.nl, Rotterdam, Netherlands). Trials with significant activity in the 500 ms pre-CS period (>7*IQR) were regarded as invalid for further analysis. Valid trials were further normalized by aligning the 500 ms pre-CS baselines and calibrating the signal so that the size of a full blink was 1. In valid normalized trials, all eyelid movements larger than 0.1 and with a latency to CR onset between 50–250 ms and a latency to CR peak of 100–250 ms (both relative to CS onset) were considered as conditioned responses (CRs). For CS only trials in the probe session we used the exact same criteria except that the latency to CR peak time was set at 100–500 ms after CS onset.

## Erasmus ladder

Mice aged 11–16 weeks were subjected to the Erasmus Ladder (Noldus, Wageningen, Netherlands). As described previously (*Vinueza Veloz et al., 2015*), the Erasmus Ladder is a fully automated system consisting of a horizontal ladder between two shelter boxes. The ladder has 2 × 37 rungs for the left and right side. Rungs are placed 15 mm apart, with alternate rungs in a descended position, so as to create an alternating stepping pattern with 30 mm gaps. All rungs are equipped with touch sensors, which are activated when subject to a pressure corresponding to more than four grams. The sensors are continuously monitored to record the position and the walking pattern of the mouse. A single crossing of the Erasmus Ladder is recorded as a trial. In this study, each mouse underwent a daily session consisting of 42 trials, for five consecutive days. Motor performance was measured by counting step durations and percentages during a trial, including short steps (steps from one high rung to the next high rung), long steps (skipping one high rung), jumps (skipping two high rungs), lower steps (a step forward steps, but the paw is placed on a low rung), back steps (a step backward steps from one high rung to the previous high rung). All data were collected and processed by ErasmusLadder 2.0 software (Noldus, Wageningen, Netherlands).

## Statistical analysis

All values are shown as mean ± s.d., unless stated otherwise. To determine means, variance and perform statistical analysis, in the electrophysiological experiments the number of cells and in the behavioral experiment the number of mice were taken as the number of replicates. Apart from the requirements for inclusion in the final datasets as stated in the separate sections for each experimental technique above, data was excluded only when the signal to noise ratio was insufficient to warrant reliable analysis. For behavioral experiments group sizes were estimated a priori using sample size calculations based on minimal relevant differences and expected variation in control cells or mice. To study compensatory eye movements the numbers are based on the VOR phase reversal. A power analysis based on repeated measures ANOVA with $\alpha = 0.05$, $\beta = 0.20$, minimal effect size f of 0.50 ($\Delta = 30°$, SD 30°, seven measurements), indicated a minimum of 11 mice per group, which were obtained (n = 11/13 for controls/mutants). For eyeblink conditioning, these numbers are based on the percentage of conditioned responses and were: $\alpha = 0.05$, $\beta = 0.20$, minimal effect size f of 0.42 ($\Delta = 25\%$, SD 30%. 15 repeats), resulting in a minimum of 14 mice per group, which were obtained (n = 15/15 for controls/mutants). For electrophysiological recordings the power analysis was based on previous experiments (*Zhou et al., 2014*), which gave a minimum group size of 10 PCs per group ($\alpha = 0.05$, $\beta = 0.20$, $\Delta = 18.1$ Hz, SD 14.0 Hz, based on Student's *t*-test) for in vitro experiments and 6 PCs per group ($\alpha = 0.05$, $\beta = 0.20$, $\Delta = 35.3$ Hz, SD 17.8 Hz) for in vivo experiments, which were obtained in all experiments (all n $\geq$ 10 for in vitro, all n $\geq$ 12 for in vivo). Inter-group comparisons were done by two-tailed Student's *t*-test. ANOVA for repeated measures was used to analyze eye movement and Erasmus ladder behavioral data; linear mixed-effect model analysis (*Boele et al., 2018*) (established in R version 1.1.442) was used to analyze eyeblink conditioning data. All statistical analyses were performed using SPSS 20.0 software. Data was considered statistically significant if p<0.05.

## Acknowledgements

We kindly thank Laura Post for mouse breeding; Nadia Khosravinia for help with behavior experiment; Yarmo Mackenbach for aid with editing the movie; Joshua J White and Dick Jaarsma for discussions and comments on the manuscript. This work was supported by an ERC starter grant (ERC-Stg #680235; MS), China Scholarship Council (#201306230130; BW), the Netherlands Organization for Scientific Research (NWO-ALW; CIDZ), the Dutch Organization for Medical Sciences (ZonMW; CIDZ), ERC-adv and ERC-POC (CIDZ), and the Center for Integrated Protein Science Munich (CIPSM; JH).

## Additional information

### Funding

| Funder | Grant reference number | Author |
|---|---|---|
| European Commission | ERC-Stg #680235 | Martijn Schonewille |
| China Scholarship Council | #201306230130 | Bin Wu |
| Nederlandse Organisatie voor Wetenschappelijk Onderzoek | ALW / Zon-Mw | Chris I De Zeeuw |
| European Commission | ERC-Adv | Chris I De Zeeuw |
| European Commission | ERC-POC | Chris I De Zeeuw |
| Center for Integrated Protein Science Munich | | Jana Hartmann |

The funders had no role in study design, data collection and interpretation, or the decision to submit the work for publication.

### Author contributions

Bin Wu, Conceptualization, Formal analysis, Supervision, Funding acquisition, Validation, Visualization, Methodology, Writing—original draft, Writing—review and editing; François GC Blot, Formal analysis, Visualization, Methodology, Writing—review and editing; Aaron Benson Wong, Data curation, Formal analysis, Visualization, Methodology, Writing—review and editing; Catarina Osório, Formal analysis, Validation, Visualization, Methodology; Youri Adolfs, Data curation, Formal analysis, Visualization, Methodology; R Jeroen Pasterkamp, Resources, Data curation, Software, Supervision, Methodology; Jana Hartmann, Esther BE Becker, Resources, Methodology, Writing—review and editing; Henk-Jan Boele, Data curation, Software, Formal analysis, Supervision, Visualization, Methodology, Writing—review and editing; Chris I De Zeeuw, Conceptualization, Resources, Software, Funding acquisition, Writing—review and editing; Martijn Schonewille, Conceptualization, Software, Supervision, Funding acquisition, Investigation, Methodology, Writing—original draft, Project administration, Writing—review and editing

### Author ORCIDs

Bin Wu (iD) https://orcid.org/0000-0003-4198-1661
Aaron Benson Wong (iD) http://orcid.org/0000-0003-1650-2710
R Jeroen Pasterkamp (iD) http://orcid.org/0000-0003-1631-6440
Esther BE Becker (iD) https://orcid.org/0000-0002-5238-4902
Chris I De Zeeuw (iD) http://orcid.org/0000-0001-5628-8187
Martijn Schonewille (iD) https://orcid.org/0000-0002-2675-1393

### Ethics

Animal experimentation: This study was performed under and all of the animals were handled according to a project license approved by the Dutch Central Committee for Animal Experiments (CCD, AVD #101002015273). Each experiment was separately verified and approved by the Animal Welfare Body (IvD/AWB, various numbers). All surgery was performed under isoflurane anesthesia combined with local anesthetics and analgesics in an effort to minimize suffering.

### Decision letter and Author response

Decision letter https://doi.org/10.7554/eLife.45590.031
Author response https://doi.org/10.7554/eLife.45590.032

# Additional files

## Supplementary files

• Supplementary file 1. Summary of the electrophysiological changes in gain- and loss-of-function TRPC3 mutants.
DOI: https://doi.org/10.7554/eLife.45590.027

• Transparent reporting form
DOI: https://doi.org/10.7554/eLife.45590.028

## Data availability

All electrophysiology and behavioral data are included in the manuscript and supporting files. Source data files have been provided for Figures 2 to 7 and Figure 2—figure supplement 1, Figure 3—figure supplement 1, Figure 4—figure supplement 1, Figure 5—figure supplement 1, Figure 6—figure supplement 1–2 and Figure 7—figure supplement 1.

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
