## [Decision Letter]

Thank you for submitting your article "TRPC3 is essential for functional heterogeneity of cerebellar Purkinje cells" for consideration by *eLife*. Your article has been reviewed by Huda Zoghbi as the Senior Editor, Jennifer Raymond as the Reviewing Editor, and three reviewers. The following individual involved in review of your submission has agreed to reveal his identity: Roy V Sillitoe (Reviewer #1).

The reviewers have discussed the reviews with one another and the Reviewing Editor has drafted this decision to help you prepare a revised submission.

Summary:

The cerebellum is organized into a series of domains that are classically defined by cell lineage, neuronal birthdate, gene expression, and afferent terminal field topography. The authors previously extended on these data by showing that cerebellar domains can also be defined as functional modules with distinct Purkinje cell firing properties. In this current manuscript, the authors go on to test whether TRPC3 molecular function is required for Purkinje cell functional heterogeneity. Using mouse genetics and electrophysiology in slice and in vivo, they demonstrate that TRPC3 indeed has an impact on Purkinje cell modules, but predominantly those defined as being zebrin II negative.

This is a very interesting paper that tackles a concept that is emerging as a central theme in cerebellar function. The results suggest that Z- and Z+ modules contribute to different behavioral functions, rather than contributing together in complementary ways to a similar or single behavior. This finding is likely to have significant impact on the fields understanding of these modules. It will greatly influence the design and direction of future experimental approaches aimed at understanding the behaviorally relevant neural signals these distinct regions of the cerebellum encode.

The data are high quality and the methods are expansive. The authors employ a powerful combination of anatomy, in vitro and in vivo physiology and behavior, and the use multiple approaches for perturbing TRPC3 including gain- and loss-of-function. Overall, this is an excellent manuscript. Addressing several concerns listed below would help to strengthen the overall interpretation of the experiments conducted, and the reviewers thought that these could be addressed in a reasonable time frame.

Essential revisions:

1) Quantification of the correlation between expression of TRPC3 and zebrin.

The anatomy is critical, and the reviewers found this to be the weakest part of the manuscript. Zebrin identity is used throughout much of the manuscript as a proxy for TRPC3 expression: Z+ is considered TRPC3- and Z- is TRPC3+. However, the evidence for these proteins' relationship is weak and based solely on non-quantified histology (Figure 1, Figure 1—figure supplement 1, Figure 2—figure supplement 1). Although the band-like expression pattern of Zebrin is clear, the expression of TRPC3 seems more homogenous, particularly in the hemispheres and anterior vermis, and there seems to be substantial TRPC3 signal in Z+ bands. The authors should consider whether the novel, and sophisticated iDISCO and light sheet microscopy approach really contributes over traditional methods. Most importantly, the relative amounts of TRPC3 and Zebrin in should be quantified.

Figure 1 D is not referenced in the text. Although the authors say that TRPC3 levels are higher in the anterior cerebellum (mainly Z-) and lower is the posterior cerebellum (mainly Z+), the immunoblots show a stronger band of TRPC3 in the posterior region. Can the authors explain? The same applies for panels D and E of Figure 2—figure supplement 1, where these immunoblots are shown again.

Much of the work done is based another indirect proxy: lobule identity. Lobule identity corresponds to average Zebrin content and not on actual Zebrin identity: lobule X → Z+ and lobules I-III → Z-. For example, all recordings of Figure 2, Figure 3, Figure 2—figure supplement 2 and Figure 3—figure supplement 1 are based on lobules X and I-III as proxies for Z+ and Z- PC modules, while trying to characterize the role of TRPC3 – so a lot rests on the claim that TRPC3 expresses in a pattern complementary to Zebrin bands. There is no figure where the amount of TRPC3 and Zebrin are quantified in the lobules used as proxies throughout the paper.

2) Some of the physiology suggests that TRPC3 affects Z+ as well as Z- regions. These results should be better acknowledged and discussed.

Figure 5: The authors say that manipulation of TRPC3 activity affected mainly Z- PCs (subsection “TRPC3 mutations selectively affect the activity in Z– olivocerebellar modules”). However, there were considerable effects in some of the Z+ PCs as well, namely in the gain of function mutants and in the tamoxifen animals. This suggests (again) that complementary expression patterns of Zebrin/TRPC3 might not be as clear as the authors state, or that there's some contamination of PC from Z+ and Z- bands. Conclusions should be adjusted accordingly

3) Some of the results suggest TRPC3 only partially accounts for the differences between Z+ and Z- functional differences; conclusions about the role of TRPC3 should be toned down accordingly.

The title would suggest that without TRPC3 you should not have Purkinje cell functional heterogeneity. This is not really the case. This is especially true since the authors say, "These results argue against a direct link between simple spike firing rate and TRPC3 levels and support…."

TRPC3 expression complements Zebrin bands in (some of) the vermis. However, in the hemispheres, where the expression of TRPC3 does not seem to follow the same pattern, there are still Z+ and Z- bands, with Purkinje cells with distinct physiological properties.

In the L7-TRPC3KO SS firing rate is decreased in Z- bands but still higher than Z+ bands (both in vitro (Figure 2) and in vivo (Figure 3), suggesting that other proteins (besides TRPC3) are playing a role.

Figure 4 A-D: The results are basically the same as in Figure 3 and Figure 3—figure supplement 1 but now recording from PCs of visually identified bands, which strengthens the results. However, this time SS firing in Z- bands without TRPC3 is as low as in Z+ bands (as desired, presumably). Do the authors think this 'discrepancy' comes from the possibility of, when recording 'blind', mixing some Z+ and Z- PCs? Perhaps the authors could try to plot the data from the two mouse lines together and check if all datapoints fall within one cloud or if some clusters emerge, highlighting potential outliers and suggesting that some of non-visually identified PCs (putative Z- and Z+) have been misidentified.

In Figure 4H we see again that SS in the absence of TRPC3 is decreased but still higher than SS in Z+ band – is this incomplete shift in SS due to a partial effect of tamoxifen or is it evidence that TRPC3 does not account for everything?

4) TRPC3 and zebrin expression should be quantified in the behaviorally relevant cerebellar regions.

Zebrin and TRPC3 protein levels must be quantified for the VOR and eyeblink regions. Otherwise, the authors are overstating the role for TRPC3 in behavior, with only indirect evidence for the amount of TRPC3 in the regions/cells relevant for the behavior. The electrophysiology done throughout the paper was not done in these regions.

The authors show no behavioral effect on VOR and a mild eyeblink conditioning deficit. Given these mostly negative results, have the authors tried, for the behavioral tests, to use the acute (tamoxifen) mutants to rule out concerns about compensatory mechanisms?

5) Validation is needed for many of the reagents, including the TRPC3 antibody, L7CreER mouse, and general characterization of the weight, locomotion, and ataxia of the mouse lines.

Using L7-Cre mice to express a transgene or create a knockout of a floxed gene is not perfect. First, retinal bipolar cells are known to express L7. Second, it has been reported that the L7-Cre mouse version used here (Barski, 2000) is more permissive than more recent strains, that is, expression of proteins under the L7 promotor have been observed anecdotally by multiple laboratories in other neurons within the cerebellum including molecular layer interneurons. Although the authors illustrate in Figure 2—figure supplement 1 that some UBCs still express TRPC3, the image suggests that there may be a significant reduction in number. It would be good to quantify this expression to ensure that other neuron types are not also effected. There is no characterization of the L7CreER mouse. What is the recombination efficiency? Is it truly selective for Purkinje cells? Finally, the authors should address the possibility that their genetic manipulation led to perturbed function of retinal bipolar cells, as these cells are also known to express TRPC3 along with L7. This is of particular importance as they use light as a visual cue within both behavioral paradigms. A control for this potential confound should be considered, or logical rationale provided for why it is not necessary.

Figure 4F. A western blot quantifying TRPC3 and really be sure the manipulation worked would be useful.

Introduction. The authors state that prior TRPC3 antibodies were of poor quality, and that the novel antibody they use from Cell Signaling is better. It is unclear why the prior antibodies were poor, and no experimental or even observational justification is provided for why this new antibody is better.

Some general characterization of the multiple mouse lines is important. including weight (the size of the animals might have an impact on the Erasmus ladder performance) and level of locomotor activity (decreased levels of locomotion could account for the mild phenotype the authors saw in the eyeblink conditioning assay (Albergaria et al., 2018). Additionally, the authors say in subsection “Animals”, that TRPC3 mutants are ataxic. Were the L7-TRPC3KO also ataxic?

---

## [Author Response]

Essential revisions:1a) Quantification of the correlation between expression of TRPC3 and zebrin.The anatomy is critical, and the reviewers found this to be the weakest part of the manuscript. Zebrin identity is used throughout much of the manuscript as a proxy for TRPC3 expression: Z+ is considered TRPC3- and Z- is TRPC3+. However, the evidence for these proteins' relationship is weak and based solely on non-quantified histology (Figure 1, Figure 1—figure supplement 1, Figure 2—figure supplement 1). Although the band-like expression pattern of Zebrin is clear, the expression of TRPC3 seems more homogenous, particularly in the hemispheres and anterior vermis, and there seems to be substantial TRPC3 signal in Z+ bands. The authors should consider whether the novel, and sophisticated iDISCO and light sheet microscopy approach really contributes over traditional methods. Most importantly, the relative amounts of TRPC3 and Zebrin in should be quantified.The reviewers are absolutely correct in stating that the expression pattern of TRPC3 is not simply complementary to that of Zebrin and that TRPC3+ is not Z- and vice versa. This is not the message we aim to convey. The immunohistochemistry (IHC) data indicate that TRPC3 is present in all Purkinje cell with a more clear banding pattern in the vermis, complementary to Zebrin, and -as the reviewers correctly note- a more homogenous expression in the hemispheres. Hence, the main message of our manuscript is that TRPC3 is a major contributor to the differences between Z+ and Z- Purkinje cells, but this is not solely based on its expression pattern. We believe that there are two contributors to the Zebrin-related specific effects of TRPC3 functioning: (1) the expression pattern of TRPC3, which is indeed mostly complementary in the vermis, but less so in the hemispheres (now quantified in the new figure 1), and (2) the expression pattern of the receptor that drives TRPC3 activation, i.e. mGluR1b, which is clearly complementary to Zebrin (Mateos, 2001). These contributing factors are now more clearly stated (see e.g., in the Discussion section). Moreover, to visualize the expression patterns more clearly, we have now re-organized Figure 1 and included a quantification of the expression pattern of TRPC3 and Zebrin (Figure 1C-D and Figure 1—figure supplement 2). This analysis helps us to visualize that the banding in the vermis is complementary. Moreover, in the hemispheres there is evidence for complementary expression in some places and for more homogenous expression of TRPC3 in others. A similar quantification was performed for the cerebellar regions controlling the behavioral tasks of eye movement adaptation and eyeblink conditioning (Figure 6—figure supplement 2B). Finally, the iDisco cleared and light-sheet imaged TRPC3 staining in the whole brain has been removed from Figure 1. We do prefer to keep this analysis as part of the manuscript, because the TRPC3 staining works better in this method. We have now annotated it and added a comparison to a cerebellar section with a Zebrin-like staining (EAAT4), side by side, to facilitate a direct comparison (Video 1).b) Figure 1 D is not referenced in the text. Although the authors say that TRPC3 levels are higher in the anterior cerebellum (mainly Z-) and lower is the posterior cerebellum (mainly Z+), the immunoblots show a stronger band of TRPC3 in the posterior region. Can the authors explain? The same applies for panels D and E of Figure 2—figure supplement 1, where these immunoblots are shown again.We understand that this was not very clear. It is important to note that the experiment in former Figure 1D was designed to determine the subcellular location of TRPC3 in both anterior and posterior cerebellar tissue and is not suited to quantitively compare TRPC3 levels between cerebellar regions. We have now added additional samples to the quantification of the western blot (Figure 2—figure supplement 1A), which confirm the previous difference. Taken together with the stainings, these results indicate that TRPC3 levels are higher in the anterior than in the posterior cerebellum. We refer to the immunoblots aimed at confirming the subcellular localization of TRPC3 in the main text (subsection “TRPC3 differentially controls the physiological properties of PCs in vitro*”*). As part of the requested additions and re-arrangement of Figure 1, the figure panel with the immunoblot has been removed there and is now only presented in Figure 2—figure supplement 1C-D.c) Much of the work done is based another indirect proxy: lobule identity. Lobule identity corresponds to average Zebrin content and not on actual Zebrin identity: lobule X → Z+ and lobules I-III → Z-. For example, all recordings of Figure 2, Figure 3, Figure 2—figure supplement 2 and Figure 3—figure supplement 1 are based on lobules X and I-III as proxies for Z+ and Z- PC modules, while trying to characterize the role of TRPC3 – so a lot rests on the claim that TRPC3 expresses in a pattern complementary to Zebrin bands. There is no figure where the amount of TRPC3 and Zebrin are quantified in the lobules used as proxies throughout the paper.

We thank the reviewers for bringing up this point; we frankly agree. The western blot results for TRPC3 depicted in the original Figure 2—figure supplement 1 were performed on tissue from lobules I-III for Z- and lobules X and part of IX for Z+. As also discussed in response to point 1a., our aim was to determine the contribution of TRPC3 to Z+ and Z– PCs and hence we chose these lobules based on the Zebrin II identity of their Purkinje cells, not their TRPC3 levels. The expression of Zebrin II in these lobules has been extensively documented (e.g. Brochu et al., 1990; Sugihara and Quy, 2007). This is now more clearly stated in the main text (subsection “TRPC3 differentially controls the physiological properties of PCs in vitro*”*) and visualized in Figure 1—figure supplement 1 and Video 1.

2a) Some of the physiology suggests that TRPC3 affects Z+ as well as Z- regions. These results should be better acknowledged and discussed

We agree; we now address the possibility that the regularity of the simple spikes and complex spikes in the Z+ Purkinje cells of the TRPC3-Mwk mice was affected by the effects of the TRPC3 mutation in UBCs and inferior olive neurons, respectively. This is now explained in the discussion (see Discussion section). In addition, we now discuss the differential consequences of TPRC3 mutations on complex spike rates in the Z- and Z+ bands in the inducible and non-inducible mice in more detail (Discussion section).

b) Figure 5: The authors say that manipulation of TRPC3 activity affected mainly Z- PCs (subsection “TRPC3 mutations selectively affect the activity in Z– olivocerebellar modules”). However, there were considerable effects in some of the Z+ PCs as well, namely in the gain of function mutants and in the tamoxifen animals. This suggests (again) that complementary expression patterns of Zebrin/TRPC3 might not be as clear as the authors state, or that there's some contamination of PC from Z+ and Z- bands. Conclusions should be adjusted accordingly

As discussed in the response to the previous comment, the effects in the gain of function mutants can be attributed directly to the global nature of this mutation, affecting other cells than Purkinje cells.

There were no changes in the parameters of the Z+ Purkinje cells in mice with tamoxifen induced deletion of TRPC3, so we assume the reviewer is referring to the effect on Z+ complex spike rate in the targeted recording experiment. This change is indeed more puzzling. We agree that the expression pattern of Zebrin and TRPC3 are not perfectly complementary, as discussed above. However, the recordings the reviewer is referring to were made in the vermis, where the complementary expression is most prominent (see new figure 1C-D). It could be that in this particular area the relative contribution of TRPC3 is larger. To make clear that the complementary pattern is not perfect we discussed the ins and outs of this issue more extensively and we modified the text accordingly (subsection “Specific expression pattern and subcellular localization of TRPC3 in the mouse brain” and the Discussion sections).

3a) Some of the results suggest TRPC3 only partially accounts for the differences between Z+ and Z- functional differences; conclusions about the role of TRPC3 should be toned down accordingly.We agree with the reviewers that TRPC3 by itself does not explain all the differences observed between Z+ and Z- Purkinje cells. This was also not the message we were aiming to convey and was the reason we included the second paragraph of the original discussion. In the new manuscript, we have made several changes to further clarify this point in the discussion (see Discussion section) and made additional changes to convey this message (see also the responses to related comment 3b, 3d and 3e below). Most importantly here, we have changed the title and last line of the abstract to more carefully describe our findings.b) The title would suggest that without TRPC3 you should not have Purkinje cell functional heterogeneity. This is not really the case. This is especially true since the authors say, "These results argue against a direct link between simple spike firing rate and TRPC3 levels and support…."

The original title stated that TRPC3 is essential for functional heterogeneity in Purkinje cells. We believe this is correct in the sense that our data indicate that TRPC3 specifically has a major influence on spiking activity in Z- Purkinje cells. The reviewer correctly states that TRPC3 by itself does not explain the complete difference between Z+ and Z- Purkinje cells. To address this, we have re-phrased our title to: “TRPC3 is a major contributor to the cellular heterogeneity of cerebellar Purkinje cells”.

c) TRPC3 expression complements Zebrin bands in (some of) the vermis. However, in the hemispheres, where the expression of TRPC3 does not seem to follow the same pattern, there are still Z+ and Z- bands, with Purkinje cells with distinct physiological properties.

We agree. In this manuscript we provide evidence that TRPC3 is a major contributor to the difference in physiological properties. The reviewer is correct in stating that other factors presumably contribute to the difference in firing rate, and probably other potential differences too.

We now more clearly state that we believe that the difference between Z-and Z+ is the result of the expression level of TRPC3, which is not completely complementary, as well as the expression pattern of its driver, mGluR1b (Mateos et al., 2001), which is completely complementary to that of Zebrin.

To address this issue we have quantified the expression of TRPC3 and Zebrin in several cerebellar regions, with new panels in Figure 1 and Figure 1—figure supplement 1 and Figure 2—figure supplement 1, as well as Figure S8B for the regions related to the tested behaviors. In addition, the factors contributing to differential TRPC3 function are now addressed in more detail in the second paragraph of the Discussion section.

*d) In the L7-TRPC3KO SS firing rate is decreased in Z- bands but still higher than Z+ bands (both* in vitro *(Figure 2) and* in vivo *(Figure 3), suggesting that other proteins (besides TRPC3) are playing a role.*

This is indeed correct, and we apologize for not stating this as clearly as needed. To make this point even more clearly, we have now included lines in the Results section, in addition to the changes made to the title, Abstract and Discussion section, mentioned above.e) Figure 4 A-D: The results are basically the same as in Figure 3 and Figure 3—figure supplement 1 but now recording from PCs of visually identified bands, which strengthens the results. However, this time SS firing in Z- bands without TRPC3 is as low as in Z+ bands (as desired, presumably). Do the authors think this 'discrepancy' comes from the possibility of, when recording 'blind', mixing some Z+ and Z- PCs? Perhaps the authors could try to plot the data from the two mouse lines together and check if all datapoints fall within one cloud or if some clusters emerge, highlighting potential outliers and suggesting that some of non-visually identified PCs (putative Z- and Z+) have been misidentified.This is an interesting point. We believe that there are two possible explanations for the different results obtained using targeted recordings compared to the blind recordings in vivo: (1) As the reviewer indicated, the recorded populations in this experiments are more ‘pure’, i.e. their Z+ or Z- identity is confirmed. This in contrast to the in vivo recordings performed blindly in lobules I-III and X, which potentially include cells with the opposite identity. However, lobule X contains virtually exclusively Z+ Purkinje cells and in lobules I-III it is estimated that >95% of the Purkinje cells is Z-. We generated the suggested plots, but found no evidence for clusters. It should be noted that (1) all firing rates in the superficial, imaging experiments reach lower levels than those in blind, deeper recordings, which probably due to the recordings conditions. Previous work demonstrated that PCs directly under exposed cerebellar surface fire simple spikes at lower rates, see Zhou et al., 2015. (2) The contribution of TRPC3 to the firing rate of Z- Purkinje cells is larger in the areas targeted using the imaging approach, the superficial parts of lobules V and VI and regions in the adjacent hemispheres. This notion is now added in subsection “TRPC3-related effects correlate with zebrin expression and are independent of development”. Taken together our results indicate that TRPC3 is responsible for a major part of the difference but cannot exclude the possibility that in some regions TRPC3 is essential for the entire difference. The fact that TRPC3 is not responsible for the entire difference is now clearly stated in the results (see 3d) and more extensively addressed in the Discussion section.f) In Figure 4H we see again that SS in the absence of TRPC3 is decreased but still higher than SS in Z+ band – is this incomplete shift in SS due to a partial effect of tamoxifen or is it evidence that TRPC3 does not account for everything?

The effects of tamoxifen were verified using immunohistochemistry and 4 weeks after injections. TRPC3 was no longer detectable in Purkinje cells. Based on this result, as well as the experiments in the normal L7-TRPC3ko mice, we conclude that TRPC3 indeed does not account for the complete difference in firing rate between Z+ and Z- Purkinje cells, at least not in all areas of the cerebellum (see 3e). To be more clear on this matter, we have adjusted the title, abstract, related Results sections and discussion (see also points 3a-b,d-e).

4a) TRPC3 and zebrin expression should be quantified in the behaviorally relevant cerebellar regions.Zebrin and TRPC3 protein levels must be quantified for the VOR and eyeblink regions. Otherwise, the authors are overstating the role for TRPC3 in behavior, with only indirect evidence for the amount of TRPC3 in the regions/cells relevant for the behavior. The electrophysiology done throughout the paper was not done in these regions.

This analysis has been performed and is now included in Figure 6—figure supplement 2B, using the same approach as in Figure 1. We can only use immunohistochemistry as both regions, but particularly the eyeblink region, are too small to analyze in western blot. In the behaviorally relevant cerebellar regions, including the flocculus, the paravermal region between lobules IV-V, VI, and the hemispheral parts of IV-V and Sim, the expression of TRPC3 is largely complementary to that of Zebrin. This is in line with what we observed in the vermis. We want to stress that the immunohistochemistry and western blot results indicate that TRPC3 is also present in Z+ PCs, but in lower quantities. This can be taken as evidence supporting the notion that it is not the amount of TRPC3 per se that is responsible for the selective effects in Z- Purkinje cells. Instead, it supports the concept of TRPC3 having a differential function based on the differences in expression levels and the differences in the pathway leading to TRPC3 activation, e.g. through mGluR1b. This is now more clearly stated in the Discussion section, molecular pathway discussed in the following paragraph.

b) The authors show no behavioral effect on VOR and a mild eyeblink conditioning deficit. Given these mostly negative results, have the authors tried, for the behavioral tests, to use the acute (tamoxifen) mutants to rule out concerns about compensatory mechanisms?

We are confident that compensatory mechanisms occur. Frankly, we have never found any mutant in which they did not occur. Compensation is part of biological systems and the more time you allow for it to occur, the more prominent it will be. One of the main points in the current study is that despite these compensatory mechanisms, there still is a difference between the Z+ controlled VOR learning (no deficit) and the Z– controlled eyeblink conditioning (mild but significant deficits despite compensation). These findings are in line with the differential expression patterns and functioning of TRPC3. We have now better explained this in the Discussion section.

5a) Validation is needed for many of the reagents, including the TRPC3 antibody, L7CreER mouse, and general characterization of the weight, locomotion, and ataxia of the mouse lines.

The TRPC3 antibody is validated by the presence of a band at the correct size in western blot and the absence of staining in Purkinje cells of the L7-TRPC3ko and the L7-TRPC3cko mice (versus the presence of the TRPC3 staining in the UBCs and inferior olive as positive control). This is now included in the first paragraph of the Results section and in the Materials and methods section. For the L7CreER mice we have now included a crossing with a reporter line to demonstrate the extent and selectivity of this line to drive Cre expression in Purkinje cells, see Figure 6—figure supplement 1F. The general characterization of both mutant mouse lines has been published before (L7-TRPC3ko: Hartmann et al., 2008; TRPC3-Mwk: Becker et al., 2009; Sekerkova et al., 2013). For the behavioral experiments we opted not to include the gain-of-function Moonwalker mice, as this is a global mutant with effects and even neurodegeneration in other cell types, resulting in a ataxic phenotype (see references above). The locomotion phenotype of the loss-of-function mutation of TRPC3 mice was tested before (in a global deletion) and only a slightly wider gait was found. Here we further expand this analysis by testing their performance on the Erasmus Ladder and found that this difference in hindlimb position did not affect their ability or speed in this locomotion task, indicating that these mice have minimal to no signs of ataxia.

b) Using L7-Cre mice to express a transgene or create a knockout of a floxed gene is not perfect. First, retinal bipolar cells are known to express L7. Second, it has been reported that the L7-Cre mouse version used here (Barski, 2000) is more permissive than more recent strains, that is, expression of proteins under the L7 promotor have been observed anecdotally by multiple laboratories in other neurons within the cerebellum including molecular layer interneurons. Although the authors illustrate in Figure 2—figure supplement 1 that some UBCs still express TRPC3, the image suggests that there may be a significant reduction in number. It would be good to quantify this expression to ensure that other neuron types are not also effected. There is no characterization of the L7CreER mouse. What is the recombination efficiency? Is it truly selective for Purkinje cells? Finally, the authors should address the possibility that their genetic manipulation led to perturbed function of retinal bipolar cells, as these cells are also known to express TRPC3 along with L7. This is of particular importance as they use light as a visual cue within both behavioral paradigms. A control for this potential confound should be considered, or logical rationale provided for why it is not necessary.

We can appreciate the concerns of the reviewers regarding the use of this L7-Cre line. Regarding the first point, the L7 promotor has indeed been reported to be expressed in retinal bipolar cells. However, as part of the analysis of compensatory eye move, we tested the optokinetic response of L7-TRPC3ko mice. Over a range of stimulus frequencies we found no difference in gain or phase, indicating that these mice do not have any visual impairments. This is in line with our experiences in numerous other mouse lines, where no effects on visual abilities were observed. Additional evidence supporting this claim could be taken from the fact that eyeblink conditioning is these mice is delayed but not completely blocked. For the ectopic expression of TRPC3 in other cerebellar cell types the only concerns are indeed the UBCs, as all other types, including molecular layer interneurons, do not express TRPC3. We understand the remark of the reviewer regarding the apparent reduction in number of TRPC3 positive UBCs in the L7-TRPC3ko mice. Visual inspection of several sections of different mice did not give any reason to assume a change. Instead, we found that in the original panel in Figure S2 the optical section was thicker for the control images than for the L7-TRPC3ko mice (this could, in hindsight, also be noticed by the intensity of the zebrin/aldolase C and calbindin images). The original image for the L7-TRPC3ko mouse in Figure 2—figure supplement 1F has been replaced by a more representative example with similar optical thickness (now in Figure 2—figure supplement 2F). It should be note that changes in UBC numbers would putatively affect eye movement adaptation more than eyeblink conditioning, hence even if there was a loss of UBCs, this could not explain the behavioral phenotype was found. The L7CreER line is remarkably efficient and selective. We have now added panel F to Figure 6—figure supplement 1, to show the results of a crossing with a reporter line and administering the same 5 injections of tamoxifen. We reach virtually complete recombination efficiency, exclusively in cerebellar Purkinje cells. This is confirmed by the absence of TRPC3 staining in the immunohistochemical analysis that was and is presented in Figure 4F).

c) Figure 4F. A western blot quantifying TRPC3 and really be sure the manipulation worked would be useful.

We agree that a western blot would make it possible to quantify the absence of TRPC3, but unfortunately the time allotted for the resubmission did not allow to breed mice, inject in adults, sacrifice four weeks later and perform the western blot. However, the results of immunohistochemical analysis of TRPC3 are in line with western blots on TRPC3 in the L7-TRPC3ko mice and the immunoblots for L7-TRPC3cko indicates that the levels of TRPC3 are similar to those in L7-TRPC3ko mice. Also, the successful deletion of TRPC3 is supported by the effect on simple spike firing rate in vivo, which is similar to that in mice with the constitutive L7-Cre.

d) Introduction. The authors state that prior TRPC3 antibodies were of poor quality, and that the novel antibody they use from Cell Signaling is better. It is unclear why the prior antibodies were poor, and no experimental or even observational justification is provided for why this new antibody is better.

The aim of this statement was to clarify why the patterned expression of TRPC3 we found could potentially previously have been missed. We agree that we cannot explain this further or do the proper comparison. As this is not directly relevant to our study, this statement has been removed and replaced by a statement on the antibody validation, as requested above.

e) Some general characterization of the multiple mouse lines is important. including weight (the size of the animals might have an impact on the Erasmus ladder performance) and level of locomotor activity (decreased levels of locomotion could account for the mild phenotype the authors saw in the eyeblink conditioning assay (Albergaria et al., 2018). Additionally, the authors say in subsection “Animals”, that TRPC3 mutants are ataxic. Were the L7-TRPC3KO also ataxic?

We agree that general features including weight and activity levels are relevant to the behavioral tests we performed. With respect to the general features, in should be noted that both lines have been previously published on: The TRPC3-Mwk mutation is featured by e.g. neurodegeneration and ataxia, and severely impacts behavior (Becker et al., 2009). Hence, we did not include these mice in any of the behavioral tests. The complete, global loss of TRPC3 has minimal to no effects on the mentioned parameters (Hartmann et al., 2008). The Purkinje cell specific knockouts (L7-TRPC3ko) mice did not show any overt changes in weight or locomotion capabilities upon visual inspection, thus we did not consistently weigh these mice. In a small sample we tested now we found no evidence for a difference in weight either (L7-TRPC3ko: 28.8 ± 5.7 g vs. controls: 27.6 ± 4.5 g; mean ± s.d., n = 3 vs. 6, adult 3-4 months old mice, equal m/f ratio). The locomotion activity of L7-TRPC3ko mice was tested in the Erasmus Ladder experiment and no significant differences were observed in the key parameters. If anything, the trend was for mutant mice to make more long steps and less short steps, which would be opposite from decreased locomotion that would lead to impaired eyeblink conditioning. This option is now included in the main text (Results section). In short, L7-TRPC3ko mice cannot be individually distinguished from control mice in any way, and as a group are only different from controls when tested in eyeblink conditioning. These remarks regarding the locomotion and weight of the mutant mice are now included in the Results section.